# Mitochondria regulate intracellular coenzyme Q transport and ferroptotic resistance via STARD7

**Soni Deshwal[1], Mashun Onishi[1], Takashi Tatsuta[1], Tim Bartsch [1], Eileen Cors[1], Katharina Ried[1], Kathrin Lemke[1], Hendrik Nolte[1], Patrick Giavalisco [1] & Thomas Langer [1,2]** ✉

Coenzyme Q (or ubiquinone) is a redox-active lipid that serves as universal electron carrier in the mitochondrial respiratory chain and antioxidant in the plasma membrane limiting lipid peroxidation and ferroptosis. Mechanisms allowing cellular coenzyme Q distribution after synthesis within mitochondria are not understood. Here we identify the cytosolic lipid transfer protein STARD7 as a critical factor of intracellular coenzyme Q transport and suppressor of ferroptosis. Dual localization of STARD7 to the intermembrane space of mitochondria and the cytosol upon cleavage by the rhomboid protease PARL ensures the synthesis of coenzyme Q in mitochondria and its transport to the plasma membrane. While mitochondrial STARD7 preserves coenzyme Q synthesis, oxidative phosphorylation function and cristae morphogenesis, cytosolic STARD7 is required for the transport of coenzyme Q to the plasma membrane and protects against ferroptosis. A coenzyme Q variant competes with phosphatidylcholine for binding to purified STARD7 in vitro. Overexpression of cytosolic STARD7 increases ferroptotic resistance of the cells, but limits coenzyme Q abundance in mitochondria and respiratory cell growth. Our findings thus demonstrate the need to coordinate coenzyme Q synthesis and cellular distribution by PARL-mediated STARD7 processing and identify PARL and STARD7 as promising targets to interfere with ferroptosis.

Coenzyme Q (ubiquinone or CoQ) is a universal multifunctional lipid with fundamental roles in cellular homeostasis[1–6]. It consists of a redox-active benzoquinone ring fused to a polyprenoid tail, whose length varies between species (ten isoprenyl units in human, nine in mouse) and which renders CoQ extremely hydrophobic. CoQ is synthesized at the mitochondrial inner membrane (IM)[1,7,8], where it serves as an electron carrier in the mitochondrial respiratory chain, shuttling electrons from respiratory complexes I and II to III (ref. [9]). As co-factor of various dehydrogenases and oxidoreductases in the IM, CoQ affects diverse processes such as pyrimidine synthesis, fatty acid oxidation or sulfide oxidation[10–12].

Cellular CoQ levels decline with age, and CoQ deficiency is associated with human disease[13]. CoQ deficiency can result from mutations in CoQ biosynthetic enzymes in rare genetic disorders[14,15] or secondarily from other cellular defects, such as oxidative phosphorylation (OXPHOS) deficiency[16–18]. Notably, although CoQ deficiency impairs respiration, some clinical presentations of CoQ-related diseases differ from those of classical mitochondrial disease. Indeed, besides its

[1]Max Planck Institute for Biology of Ageing, Cologne, Germany. [2]Cologne Excellence Cluster on Cellular Stress Responses in Aging-Associated Diseases (CECAD), University of Cologne, Cologne, Germany. ✉e-mail: tlanger@age.mpg.de

metabolic and bioenergetic functions in mitochondria, CoQ acts as a lipophilic antioxidant and supports enzymatic reactions in other cellular membranes[19–21]. Suppression of lipid peroxidation by CoQ and the NAD(P)H-dependent oxidoreductase FSP1 (previously known as AIFM2) in the plasma membrane (PM) protects cells against ferroptosis[19,20]. Ferroptosis is an iron-dependent cell death pathway that is associated with various degenerative disorders and that has been recognized as a promising anti-cancer strategy[22–29]. Despite the emerging importance of non-mitochondrial CoQ for cellular homeostasis, however, it remains unknown how extremely hydrophobic CoQ molecules are distributed to other cellular membranes after their synthesis in mitochondria.

The neuronal loss of the mitochondrial rhomboid protease PARL causes respiratory complex III defects and Leigh-like neurodegeneration, accompanied by CoQ deficiency[30]. CoQ loss precedes OXPHOS defects upon ablation of *Parl*, suggesting that PARL may regulate CoQ accumulation directly[30]. PARL is a member of the rhomboid family of intramembrane cleaving peptidases with pleiotropic roles in mitochondria[31]. Proteomic substrate profiling in cultured cells identified various PARL substrate proteins, many of them associated with human disease[32–36]. PARL regulates PINK1-dependent mitophagy[37,38], predisposes cells to apoptosis cleaving Smac/DIABLO[32] and maintains respiratory complex III activity by processing of the quality surveillance factor TTC19 (ref. [39]). PARL-mediated cleavage of STARD7, a START-domain-containing lipid transfer protein for phosphatidylcholine (PC), allows its partitioning between the intermembrane space (IMS) and the cytosol[40]. PC transport across the IMS by mitochondrial STARD7 preserves respiration and cristae morphogenesis, but the function of cytosolic STARD7 remained enigmatic[40].

In this Article, we demonstrate that PARL regulates both the synthesis and cellular distribution of CoQ via STARD7. While mitochondrial STARD7 ensures CoQ synthesis, cytosolic STARD7 is required for CoQ transport to the PM and for the protection of cells against ferroptosis.

## Results

### CoQ biosynthesis depends on STARD7 processing by PARL

We determined the proteome of WT and *PARL*−/− mice brain homogenates and found many proteins involved in CoQ biosynthesis reduced in the absence of PARL, while OXPHOS subunits were not affected (Fig. 1a, Extended Data Fig. 1a and Supplementary Table 1). The levels of both CoQ10 and CoQ9, differing in the length of the polyisoprenoid lipid tail, were decreased in *PARL*−/− brain and heart (Fig. 1b and Extended Data Fig. 1b). The former is the major form of CoQ in humans while the latter is predominant in mouse (hereafter we refer to both together as CoQ unless stated otherwise)[1,7]. To define how Parl affects CoQ abundance, we performed $^{13}C_6$-glucose tracing experiments and observed an impaired CoQ synthesis from $^{13}C_6$-glucose in PARL-deficient cells (Fig. 1c). Both CoQ9 and CoQ10 levels were restored upon expression of PARL in *PARL*−/− cells, but not in the presence of its proteolytically inactive variant PARL$^{S277A}$, demonstrating that the synthesis of CoQ depends on PARL-mediated proteolysis (Fig. 1d,e and Extended Data Fig. 1c).

To identify protein(s) whose impaired proteolysis may affect the CoQ synthesis in the absence of PARL, we determined CoQ levels in cells lacking known PARL substrates, namely STARD7, PGAM5, SMAC/DIABLO, CLPB and PINK1 (Fig. 1f and Extended Data Fig. 1d). As CoQ levels are not affected in *Ttc19*−/− mice, we excluded this substrate of PARL from our analysis[30]. We observed significantly decreased CoQ levels in *STARD7*−/− cells, which phenocopied *PARL*−/− cells (Fig. 1f).

PARL cleavage of the PC transfer protein STARD7 during mitochondrial import allows partitioning of STARD7 between the IMS and the cytosol (Fig. 1g)[40]. To understand how the compartmentalization of STARD7 affects CoQ levels, we expressed in *STARD7*−/− cells STARD7 variants, which specifically accumulated in the cytosol or within mitochondria (cyto-STARD7 and mito-STARD7; Fig. 1h and Extended Data Fig. 1e). The localization of STARD7 can be restricted to mitochondria by replacing the 77 amino acids of STARD7 by the N-terminal region of MICU1, which is cleaved off by the mitochondrial intermediate peptidase IMMP1L and results in the accumulation of unaltered, mature STARD7 exclusively in the IMS[40]. Notably, unprocessed STARD7 is present in the IM of *PARL*−/− cells[40] but is not sufficient to ensure the accumulation of CoQ, suggesting that STARD7 activity depends on its maturation. Expression of mito-STARD7 but not cyto-STARD7 (lacking the mitochondrial targeting sequence) restored CoQ9 and CoQ10 levels in *STARD7*−/− cells (Fig. 1i and Extended Data Fig. 1f), consistent with the requirement of mito-STARD7 for mitochondrial functions[40]. Maintenance of CoQ levels depended on the PC transport activity of mito-STARD7. Expression of the mutant STARD7$^{R189Q}$, which does not bind PC[41], did not suppress the CoQ deficiency of *STARD7*−/− cells (Fig. 1j and Extended Data Fig. 1g).

In further experiments, we examined whether *PARL*−/− cells contain lower CoQ levels due to lack of mito-STARD7. Strikingly, expression of mito-STARD7 but not cyto-STARD7 restored CoQ levels, basal and maximal respiration, and ATP production in *PARL*−/− cells (Fig. 1k–n and Extended Data Fig. 1h,i). We therefore conclude that PARL maintains CoQ synthesis by proteolytic cleavage of STARD7 and its accumulation in the IMS.

### PARL and STARD7 protect cells against ferroptosis

CoQ serves as an antioxidant in the PM, where the oxidoreductase FSP1 preserves the reduced CoQ pool, limiting lipid peroxidation and ferroptosis[19,20]. The FSP1-CoQ pathway has been identified as a second ferroptosis suppressive mechanism, which acts in parallel to glutathione peroxidase GPX4 and which renders some cancer cell lines refractory to ferroptosis upon GPX4 inhibition[19,20]. We therefore reasoned that decreased CoQ synthesis may increase the susceptibility of *PARL*−/− or *STARD7*−/− cells towards ferroptosis. Analyses of data from the Cancer

**Fig. 1 | PARL regulates CoQ levels via STARD7 processing. a**, Volcano plot of WT and *Parl*−/− brain proteomes of 5-week-old male mice. Proteins significantly altered between WT and *PARL*−/− cells are highlighted in red. **b**, CoQ9 levels were determined by MS in brain and heart homogenates of 5-week-old male WT and *PARL*−/− (*n* = 9 WT and *n* = 5 *PARL*−/− animals) mice. **c**, CoQ synthesis traced with $^{13}C_6$ glucose in WT and *PARL*−/− HeLa cells. **d,e**, Total CoQ levels in WT and *PARL*−/− HeLa cells complemented with either PARL or PARL$^{S277A}$. **f**, CoQ abundance in HeLa cells lacking the indicated PARL substrate proteins (*n* = 10 WT, *n* = 4 *STARD7*−/−, *PARL*−/−, *PGAM5*−/−, *CLPB*−/−, *n* = 12 *SMAC*−/−, *n* = 11 *PINK1*−/− biologically independent samples). **g**, PARL cleavage of STARD7 during mitochondrial import allows localization of STARD7 to both IMS and cytosol (image created with BioRender.com). OM, mitochondrial outer membrane. **h**, Domain structure of STARD7 and variants accumulating exclusively in the IMS (mito-STARD7) and the cytosol (cyto-STARD7). The mitochondrial targeting sequence (MTS) derived from MICU1 (amino acids 1–60) is cleaved off by IMMP1L protease, generating identical mature STARD7 in the IMS. cyto-STARD7 lacks the MTS (amino acids 1–76) of STARD7 (image created with BioRender.com). TM: transmembrane domain **i,j**, CoQ levels determined by MS in *STARD7*−/− cells expressing mito-STARD7 or cyto-STARD7 (*n* = 8 WT, *STARD7*−/− and *STARD7*−/− complemented with cyto-STARD7, *n* = 7 *STARD7*−/− complemented with mito-STARD7 biologically independent samples) (**i**) or mito-STARD7$^{R189Q}$ deficient in PC binding (**j**). **k–n**, CoQ abundance (**k**), basal respiration (**l**), maximal respiration (**m**) and ATP production (**n**) in WT and *PARL*−/− cells expressing either mito-STARD7 or cyto-STARD7. Data were analysed by Instant Clue software and are represented by 95% confidence (**b–f,i–n**) interval for the mean. In **a,c–e,j–k**, *n* = 5 biologically independent samples. In **l–n**, *n* = 6 WT, *PARL*−/− and *n* = 5 *PARL*−/− complemented with mito- and cyto-STARD7 biologically independent samples. In **b–f,i–n**, the *P* values were calculated using a two-tailed Student's *t*-test for unpaired comparisons. In **b,d,e,i**, The central band of each box is the 50% quantile, and the box defines the 25% (lower) and 75% (higher) quantile. The whiskers represent the minimum and maximum value in the data. Source numerical data are available in source data.

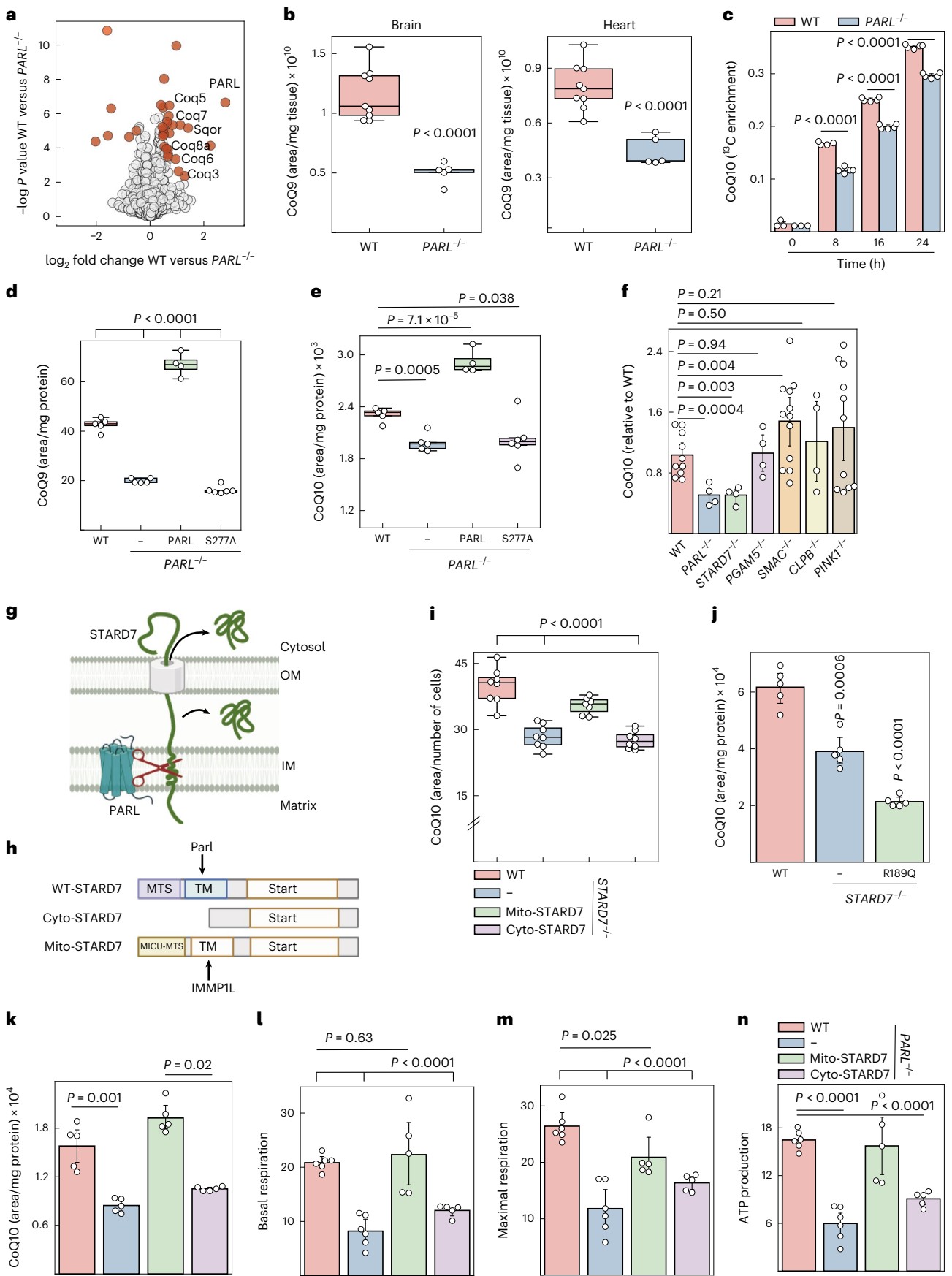

Therapeutics Response Portal (CTRP), which correlates data on gene expression and drug resistance for over 800 cancer cell lines, revealed that, similar to FSP1 (refs. [19,20]), the expression of PARL and STARD7 positively correlates with the resistance of cancer cells to GPX4 inhibitors, such as RSL3, ML210 and ML162 (Fig. 2a). Other substrates of PARL did not show this correlation, indicating that PARL may counteract ferroptosis via STARD7.

To corroborate these findings experimentally, we assessed the susceptibility of *PARL*[−/−] and *STARD7*[−/−] cells to ferroptosis after inhibition of the GPX4-dependent cellular defence mechanism (Fig. 2b). To this end, we induced ferroptosis by inhibiting either GPX4 or cystine–glutamate antiporter system $X_c^-$ (SLC7A11) with erastin and, when indicated, incubated cells in addition with polyunsaturated fatty acids (PUFAs), such as arachidonic acid 20:4, conferring additional oxidative stress to cellular membranes (Fig. 2b)[21,42]. Erastin treatment induced ferroptosis in a dose-dependent manner (Extended Data Fig. 2a). A combination of erastin and PUFA treatment boosted ferroptosis, which can be inhibited by the antioxidant ferrostatin-1 (Fer1) or by supplementing cells with synthetic, soluble CoQ2, but not by the pan-caspase inhibitor QvD (Fig. 2c and Extended Data Fig. 2b). Both *PARL*[−/−] and *STARD7*[−/−] cells exhibited an increased susceptibility to ferroptosis induced by treating cells with erastin and PUFAs (Fig. 2c and Extended Data Fig. 2c), or with the GPX4 inhibitor RSL3 (Fig. 2d). Cultivating cells in the presence of Fer1 or CoQ2 suppressed cell death under these conditions. We also deleted *PARL* and *STARD7* in the colon cancer cell line HCT116 (Fig. 2g,h and Extended Data Fig. 2d,e). Similar to HeLa cells, CoQ10 and CoQ9 levels were decreased in isolated monoclones lacking PARL or STARD7 (Fig. 2e,f), which were susceptible to erastin-induced ferroptosis (Fig. 2g,h). Thus, both PARL and STARD7 protect cells against ferroptosis upon inhibition of GPX4.

### Dual localization of STARD7 preserves ferroptotic resistance

The CoQ deficiency in *PARL*[−/−] and *STARD7*[−/−] cells might explain the increase in their ferroptotic vulnerability when GPX4 is inhibited. We therefore assessed the resistance of *STARD7*[−/−] cells expressing STARD7, mito-STARD7 or cyto-STARD7 against ferroptosis that is induced by the combined treatment with erastin and PUFAs. Re-expression of STARD7 ensured the survival of STARD7-deficient cells under these conditions, substantiating the requirement of STARD7 for protection against ferroptosis (Fig. 2i,j). Expression of cyto-STARD7 did not affect the survival of *STARD7*[−/−] cells (Fig. 2i,j), most likely due to persistent CoQ deficiency in these cells (Fig. 1i). CoQ deficiency would impair ferroptosis defence in both the PM and the mitochondria via FSP1 and dihydroorotate dehydrogenase (DHODH), respectively[19,20,26]. Unexpectedly, however, cells harbouring mito-STARD7 exhibited the same sensitivity to ferroptosis as *STARD7*[−/−] cells (Fig. 2i,j), although mito-STARD7 preserves total CoQ levels (Fig. 1i). These experiments demonstrate that the ferroptotic vulnerability of *STARD7*[−/−] cells cannot be explained solely by the impaired CoQ synthesis in these cells. Rather, suppression of ferroptosis depends on STARD7 being present both in the IMS and the cytosol.

### Cytosolic STARD7 is a suppressor of ferroptosis

Cyto-STARD7 is dispensable for CoQ synthesis but required for suppression of ferroptosis even in cells that harbour STARD7 in the IMS and maintain CoQ synthesis. To examine whether cyto-STARD7 limits the cellular resistance against ferroptosis, we overexpressed cyto-STARD7 in wild-type (WT) HeLa cells. We isolated three cell lines overexpressing cyto-STARD7 at different levels (Extended Data Fig. 3a,b) and monitored their susceptibility to ferroptosis induced by erastin, erastin and PUFAs, or RSL3 (Fig. 3a–d). Increased STARD7 levels in the cytosol promoted cell survival after 24 h, when WT cells underwent ferroptosis, which was suppressed in the presence of Fer1 or CoQ2 but not in the presence of the pan-caspase inhibitor QVD (Fig. 3a–d and Extended Data Fig. 3c). A kinetic analysis pointed towards a dose-dependent response, with ferroptosis occurring first in cells expressing the lowest levels of cyto-STARD7 (Fig. 3b).

It is conceivable that cyto-STARD7 expression increases CoQ synthesis, indirectly affecting ferroptotic resistance of the cells. However, we did not observe altered CoQ levels nor increased CoQ synthesis in $^{13}C_6$-glucose tracing experiments upon overexpression of cyto-STARD7 (Extended Data Fig. 3d,e).

Together, these experiments demonstrate that the CoQ-dependent cellular resistance against ferroptosis depends on cytosolic STARD7, whose protein level limits ferroptosis induced by various stimuli.

### STARD7 supports the FSP1-CoQ defence pathway

The identification of cyto-STARD7 as a ferroptosis suppressor acting independent of GPX4 suggests that cyto-STARD7 may promote the transport of CoQ from mitochondria to the PM, increasing the availability of CoQ and ferroptotic resistance. We therefore examined whether the suppressive effect of cyto-STARD7 against ferroptosis depends on FSP1 in the PM. We induced ferroptosis in WT cells and cells expressing cyto-STARD7 and treated the cells concomitantly with the FSP1 inhibitor iFSP1. FSP1 inhibition blunted the protective effect of cyto-STARD7 against ferroptosis induced with RSL3 (Fig. 3e), erastin (Fig. 3f,g) or with erastin and PUFAs (Extended Data Fig. 3f).

To confirm that CoQ is required for the protective function of STARD7 against ferroptosis, we inhibited CoQ synthesis with 4-carboxybenzaldehyde (4-CBA), targeting the CoQ biosynthetic enzyme CoQ2 (ref. [43]). 4-CBA treatment decreased cellular CoQ10 and CoQ9 levels in a dose-dependent manner (Fig. 3h,i) and abolished the protective effect of cyto-STARD7 against RSL3-induced ferroptosis (Fig. 3j).

Together, these experiments demonstrate that cyto-STARD7 protects cells against ferroptosis via the FSP1-CoQ pathway in the PM.

### CoQ transport from mitochondria requires cytosolic STARD7

We then performed cellular fractionation experiments to directly monitor CoQ levels in the PM and in mitochondrial membranes (Fig. 4a). A proteomic analysis of the cellular fractions obtained by differential centrifugation revealed the expected strong enrichment of mitochondrial proteins (MitoCarta 3.0) in fraction 1 (isolated by centrifugation

---

**Fig. 2 | PARL and STARD7 preserve cells against ferroptosis. a**, Correlation of high expression of PARL and STARD7 with the resistance of cancer cell lines to the GPX4 inhibitors ML210, ML162 and RSL3. Data were mined from CTRP and show z-scores of Pearson's correlation coefficients. The central band of each box is the 50% quantile, and the box defines the 25% (lower) and 75% (higher) quantile. The whiskers represent the minimum and maximum value in the data, and outliers are indicated by a plus sign (greater distance than 1.8 times interquartile range away from the median). **b**, Scheme illustrating FSP1-CoQ- and GPX4-dependent oxidative defence pathways as two independent mechanisms protecting against lipid peroxidation and ferroptosis (image created with BioRender.com). **c,d**, Ferroptosis was induced in WT, *PARL*[−/−] and *STARD7*[−/−] HeLa cells with either erastin (3 µM) + PUFA (arachidonic acid 20:4, 40 µM) (n = 3 biologically independent experiments) (c) or RSL3 (200 nM) (d) in the presence of ferrostatin-1 (Fer1, 1 µM) and CoQ2 (1 µM) as indicated, and cell death was monitored after 24 h (n = 2). **e,f**, Total CoQ10/9 levels in WT, *PARL*[−/−] and *STARD7*[−/−] HCT116 cells. #1 and #2 represent two different clones of indicated genotype. **g,h**, Increased ferroptotic vulnerability of two different clones of *PARL*[−/−] and *STARD7*[−/−] HCT116 cells compared with WT upon treatment with indicated concentrations of erastin for 24 h. **i**, Cell death in WT, *STARD7*[−/−] and in *STARD7*[−/−] cells expressing mito-STARD7, cyto-STARD7 or STARD7 after 9 h in the presence of the indicated compounds. **j**, Representative images from **i** showing dead cells in magenta and living cells with phase contrast after 9 h. In **c–i**, data were analysed by Instant Clue software and are represented by 95% confidence interval of the mean. n = 3 (**d,i**), n = 5 (**e,f**) and n = 4 (**g,h**) biologically independent experiments. In **c–i**, the P values were calculated using a two-tailed Student's t-test for unpaired comparisons. Source numerical data are available in source data.

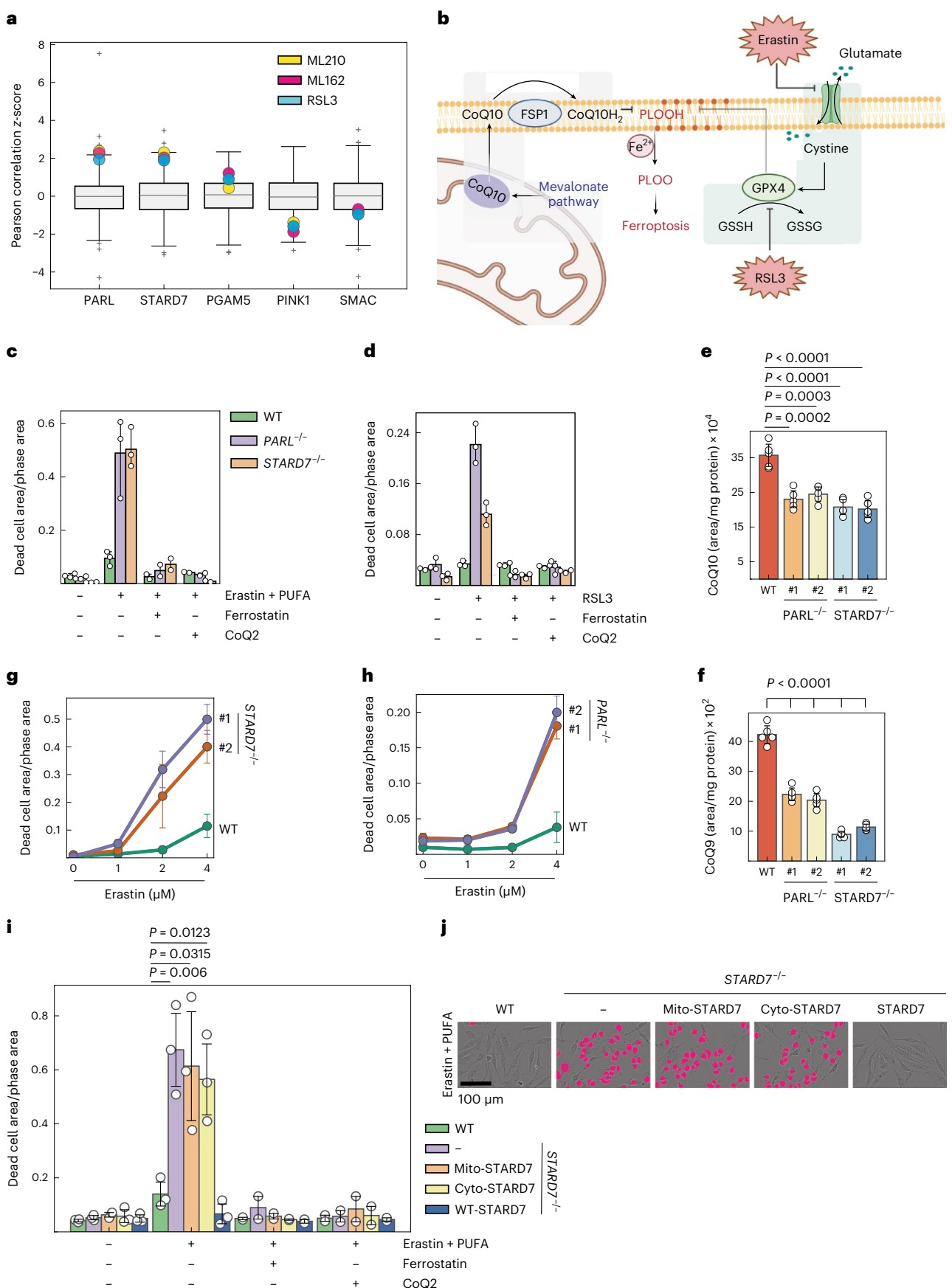

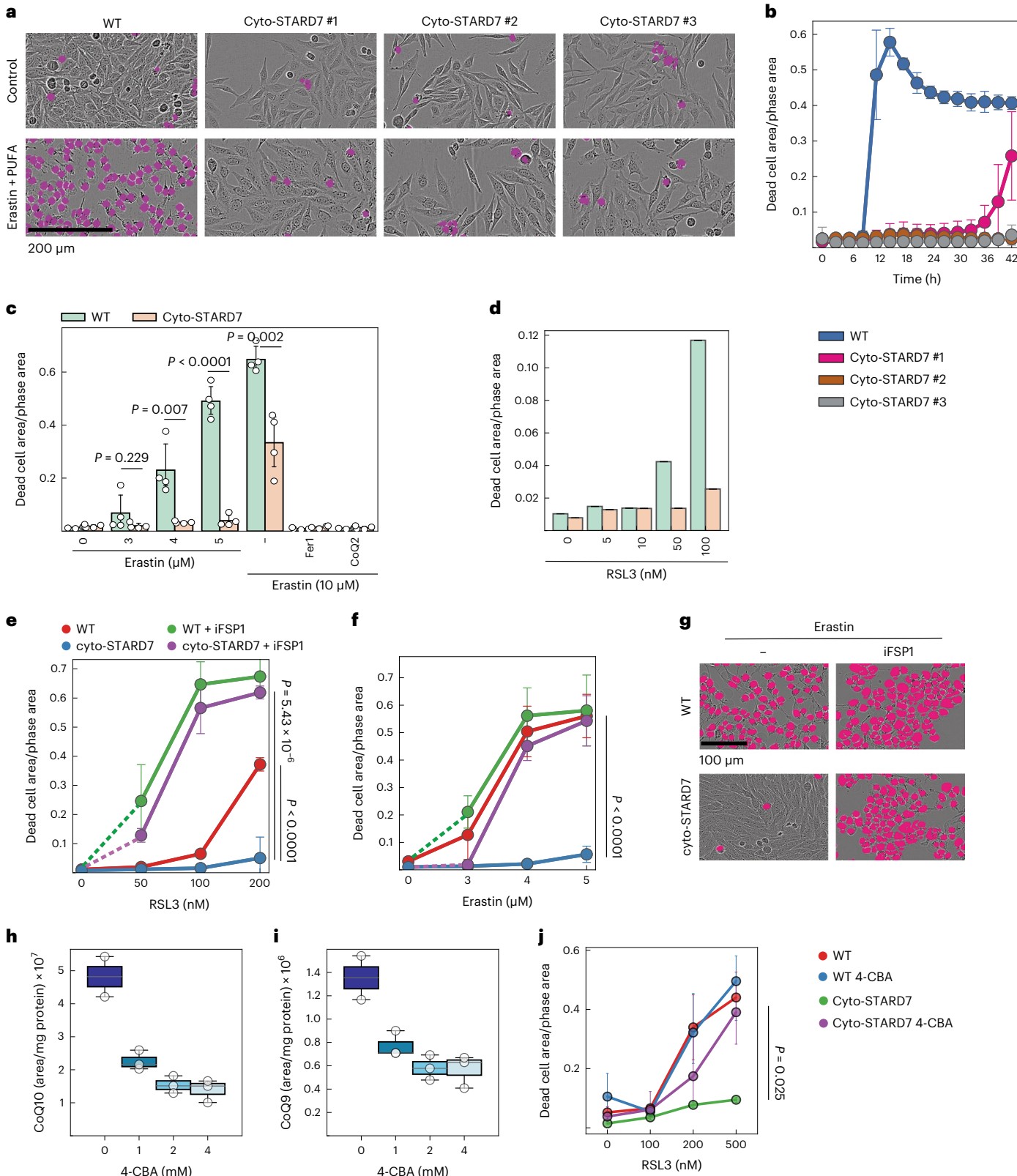

at 8,000*g*) (Fig. 4b,c, Extended Data Fig. 4a and Supplementary Table 2). On the other hand, PM proteins were more broadly distributed among the different fractions but enriched in fraction 2 (isolated by centrifugation at 12,000*g*). However, fraction 3 (isolated by centrifugation at 40,000*g*) contained the lowest number of mitochondrial proteins, further minimizing cross-contamination of the PM fraction with mitochondrial membranes that contain higher concentrations of

CoQ than the PM[44]. We therefore used this PM fraction to assess how STARD7 affects the relative distribution of CoQ levels between the PM and mitochondrial membranes.

The majority of CoQ10 and CoQ9 (~90%) was present in mitochondrial membranes, as expected (Fig. 4d,e). To monitor the effect of STARD7 on the cellular distribution of CoQ, we determined CoQ levels in the mitochondrial membrane fraction (fraction 1) and the

**Fig. 3 | Cyto-STARD7 suppresses ferroptosis via CoQ-FSP1 pathway.**
**a,b**, Ferroptosis was induced in WT and three different clones of cyto-STARD7 overexpressing cells with erastin (3 μM) in the presence of PUFA (arachidonic acid 20:4, 40 μM), and cell death was monitored. Representative images from **b** after 18 h (**a**) and kinetics of cell death (**b**) are shown. No cell death was observed in WT and cyto-STARD7 overexpressing cells treated with DMSO. **c,d**, Cell death measured in WT and cyto-STARD7-overexpressing cells with different ferroptosis inducers, including erastin (**c**) and GPX4 inhibitor RSL3 (**d**) after 24 h. Ferrostatin-1 (Fer1, 1 μM) and CoQ2 (1 μM) were used to inhibit ferroptosis. **e–g**, Ferroptosis was induced by blocking either only GPX4 arm via RSL3 (**e** and **g**) or erastin (**f**) or both GPX4 and FSP1-CoQ arm via combination of erastin/GPX4 and FSP1 inhibitor iFSP1 (5 μM) in WT and cyto-STARD7-overexpressing cells. In **g**, representative images from **f** show dead cells in magenta and alive cells in phase contrast after 24 h of treatment. **h,i**, CoQ10/9 levels were measured

in HeLa cells treated with either DMSO (*n* = 2) or indicated concentrations of coenzyme Q2 (CoQ2, polyprenyltransferase) inhibitor 4-CBA for 48 h (*n* = 3). The central band of each box is the 50% quantile, and the box defines the 25% (lower) and 75% (upper) quantile. The whiskers represent the minimum and maximum value in the data. Source numerical data are available in source data. **j**, To inhibit CoQ synthesis, WT and cyto-STARD7-overexpressing cells were treated with 4-CBA (2 mM). After 24 h of treatment, ferroptosis was induced by indicated concentrations of RSL3, and cell death was measured at 24 h. In **b**, **c**, **e**, **f** and **h–j**, data were analysed by Instant Clue software and are represented by 95% confidence interval of the mean. In **d**, data were analysed by Instant Clue software and are represented by the average of *n* = 2 biologically independent experiments. *n* = 3 (**b**, **e** and **f**) and *n* = 4 (**c** and **j**) biologically independent experiments. In **c**, **e**, **f** and **j**, the *P* values were calculated using a two-tailed Student's *t*-test for unpaired comparisons.

PM fraction (fraction 3), which were isolated from WT and *STARD7⁻/⁻* HeLa cells and from *STARD7⁻/⁻* HeLa cells expressing mito-STARD7 or cyto-STARD7 (Fig. 4f). The loss of STARD7 was associated with decreased CoQ9 and CoQ10 levels in mitochondria and the PM (Fig. 4f), rationalizing the increased susceptibility of these cells to ferroptosis. Expression of mito-STARD7 in *STARD7⁻/⁻* cells restored CoQ levels in mitochondria but not or to a lesser extent in the PM (Fig. 4f). Expression of cyto-STARD7 did not alter CoQ levels in either membrane relative to *STARD7⁻/⁻* cells (Fig. 4f). These results are consistent with the requirement of mito-STARD7 for CoQ synthesis and explain the increased ferroptotic vulnerability of cells lacking cyto-STARD7 by decreased CoQ levels in the PM. We therefore conclude that mito-STARD7 is sufficient to maintain CoQ synthesis in mitochondria, whereas cyto-STARD7 preserves the CoQ pool in the PM, facilitating CoQ transport from mitochondria.

We substantiated these experiments determining CoQ levels in mitochondrial membranes (fraction 1) and the PM (fraction 3) of WT cells overexpressing cyto-STARD7 (Fig. 4g and Extended Data Fig. 4b). CoQ9 and CoQ10 levels were significantly increased in the PM (fraction 3) of these cells. On the other hand, mitochondrial CoQ10, but not CoQ9, was decreased in cells expressing cyto-STARD7 (Fig. 4h). These data corroborate the critical role of cyto-STARD7 for cellular CoQ distribution and reveal that cyto-STARD7 limits CoQ accumulation in the PM.

### CoQ interaction with STARD7 in vitro
To gain further insight into the role of STARD7 for CoQ transport to the PM, we examined a possible direct interaction of CoQ with STARD7 in vitro (Fig. 5a). STARD7 was purified after expression in *Escherichia coli* and incubated with liposomes containing CoQ variants differing in the length of their polyprenoid tail. After re-isolation of STARD7, we determined by mass spectrometry (MS) STARD7-associated CoQ variants that were extracted from liposomes (Fig. 5b). We detected CoQ4 in association with STARD7 but not in control samples, whereas the more hydrophobic variants CoQ9 and CoQ10 were not recovered. It is conceivable that their high hydrophobicity precludes their membrane extraction or their co-purification with STARD7 in vitro. It should be noted that liposomes did not contain PC in these experiments, which is the known substrate of STARD7 (ref. [45]). Increasing the PC

concentration in liposomes indeed allowed the co-purification of an increasing amount of PC with STARD7, which was accompanied by decreased binding of CoQ4 to STARD7 (Fig. 5c). Thus, PC competes with CoQ4 for STARD7 binding. Consistently, mutating R189 within the lipid binding groove of STARD7, which was found to abolish PC binding[41] (Fig. 5d,e), also impaired binding of CoQ4 (Fig. 5f).

Together, we conclude from these experiments that STARD7 can bind to CoQ4 in vitro, suggesting that it directly affects CoQ transport from mitochondria to the PM.

### Overexpression of cyto-STARD7 limits respiratory cell growth
The cyto-STARD7-dependent cellular CoQ distribution points to a regulatory role of PARL, whose proteolytic activity determines the relative distribution of STARD7 between mitochondria and the cytosol. While overexpression of cyto-STARD7 increases CoQ in the PM and protects against ferroptosis, it decreases CoQ levels in the mitochondria and thereby may affect mitochondrial functions. We therefore examined how cyto-STARD7 overexpression affects the growth and respiratory competence of the cells. Expression of cyto-STARD7 did not affect cell growth in the presence of glucose (Fig. 6a) and had very limited effects on oxygen consumption and ATP production of the cells (Fig. 6b). In contrast, cells harbouring increased levels of cyto-STARD7 grew significantly slower under respiring conditions on galactose-containing medium (Fig. 6c). We observed significantly reduced basal and maximal respiration and ATP production, if cells overexpressing cyto-STARD7 were grown on galactose-containing medium (Fig. 6d). We did not observe any changes in the protein expression of OXPHOS subunits or CoQ biosynthetic machinery (Extended Data Fig. 5a). Thus, increased levels of cyto-STARD7 (relative to mito-STARD7) confer ferroptotic resistance by increasing CoQ levels in the PM, but limit respiratory cell growth. These findings highlight the importance of regulating the relative accumulation of STARD7 in both compartments, the mitochondria and the cytosol.

## Discussion
We demonstrate that the lipid transfer protein STARD7 is required for both the synthesis of CoQ within mitochondria and for CoQ transport from mitochondria to the PM (Fig. 6d). PARL cleavage of STARD7 ensures the dual localization of STARD7 to the mitochondrial IMS and

**Fig. 4 | Cyto-STARD7 is required for CoQ export from the mitochondria.**
**a**, Scheme showing the fractionation protocol for HeLa cells to isolate mitochondria and PM fractions with differential centrifugations (image created with BioRender.com). **b,c**, Heat maps showing the distribution of mitochondrial (**b**) and plasma membrane (**c**) proteins in different fractions of cells determined by MS. #1, 8,000*g* pellet; #2, 12,000*g* pellet; #3, 40,000*g* pellet and #4, 100,000*g* pellet. *n* = 5 biologically independent experiments. **d,e**, Total levels of CoQ10/9 measured in indicated fractions of HeLa cells. CoQ distribution within the cell is indicated in per cent in each fraction. **f,g**, Differences in the abundance of

CoQ10, CoQ9, PC, PE (**f,g**) and sphingomyelin (SM) (**g**) in mitochondrial and plasma membrane fractions relative to WT in *STARD7⁻/⁻* cells and *STARD7⁻/⁻* cells complemented with either mito-STARD7 or cyto-STARD7 (**f**) and in WT cells overexpressing cyto-STARD7 (**g**). In **d–g**, data were analysed by Instant Clue software and are represented by 95% confidence interval of the mean. *n* = 4 (**d,e,g**) and *n* = 5 (**f**) biologically independent experiments. In **d–g**, the *P* values were calculated using a two-tailed Student's *t*-test for unpaired comparisons (**d,e**) and two-tailed one-sample *t*-test for unpaired comparison (**f** and **g**). Source numerical data are available in source data.

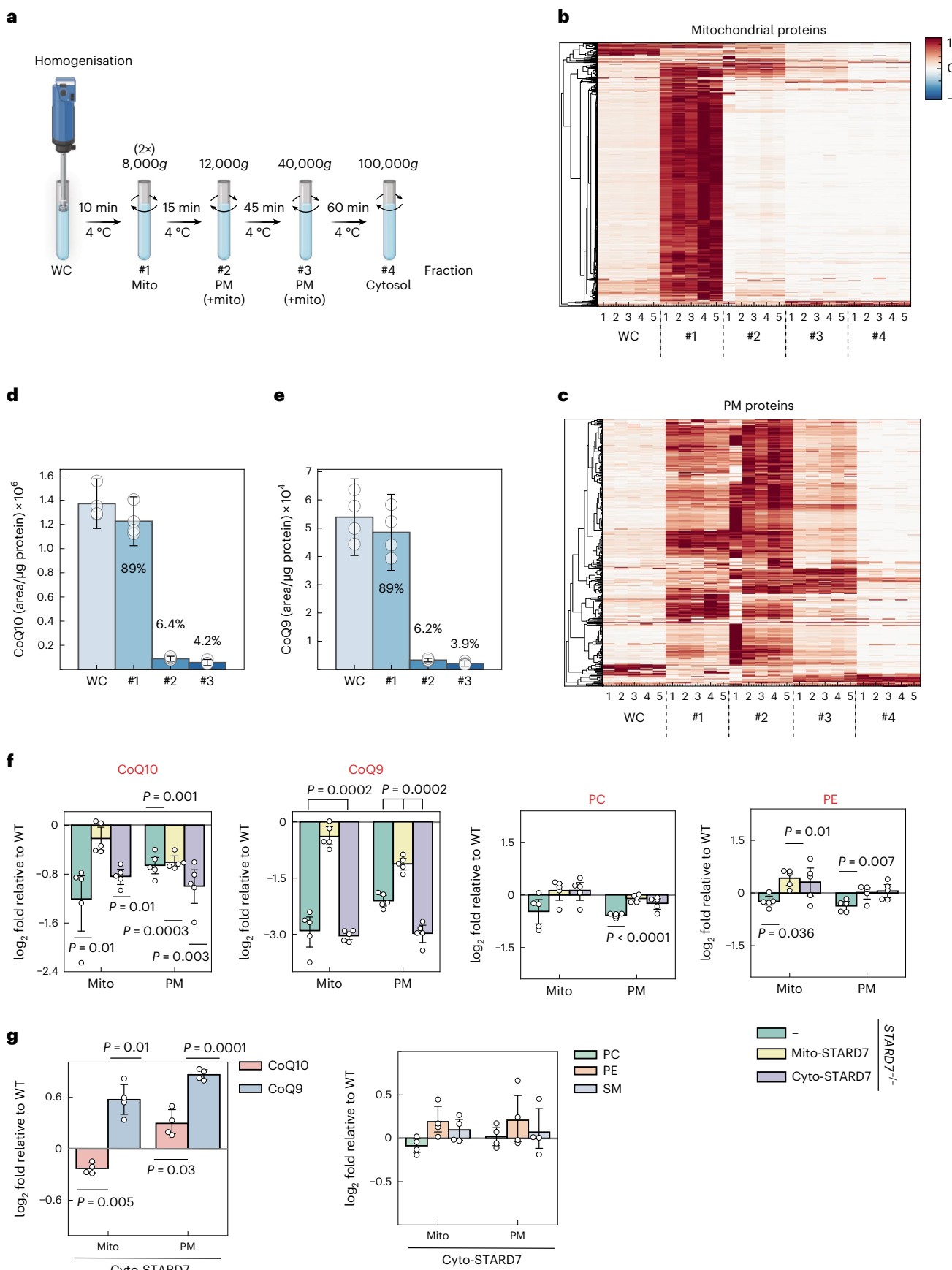

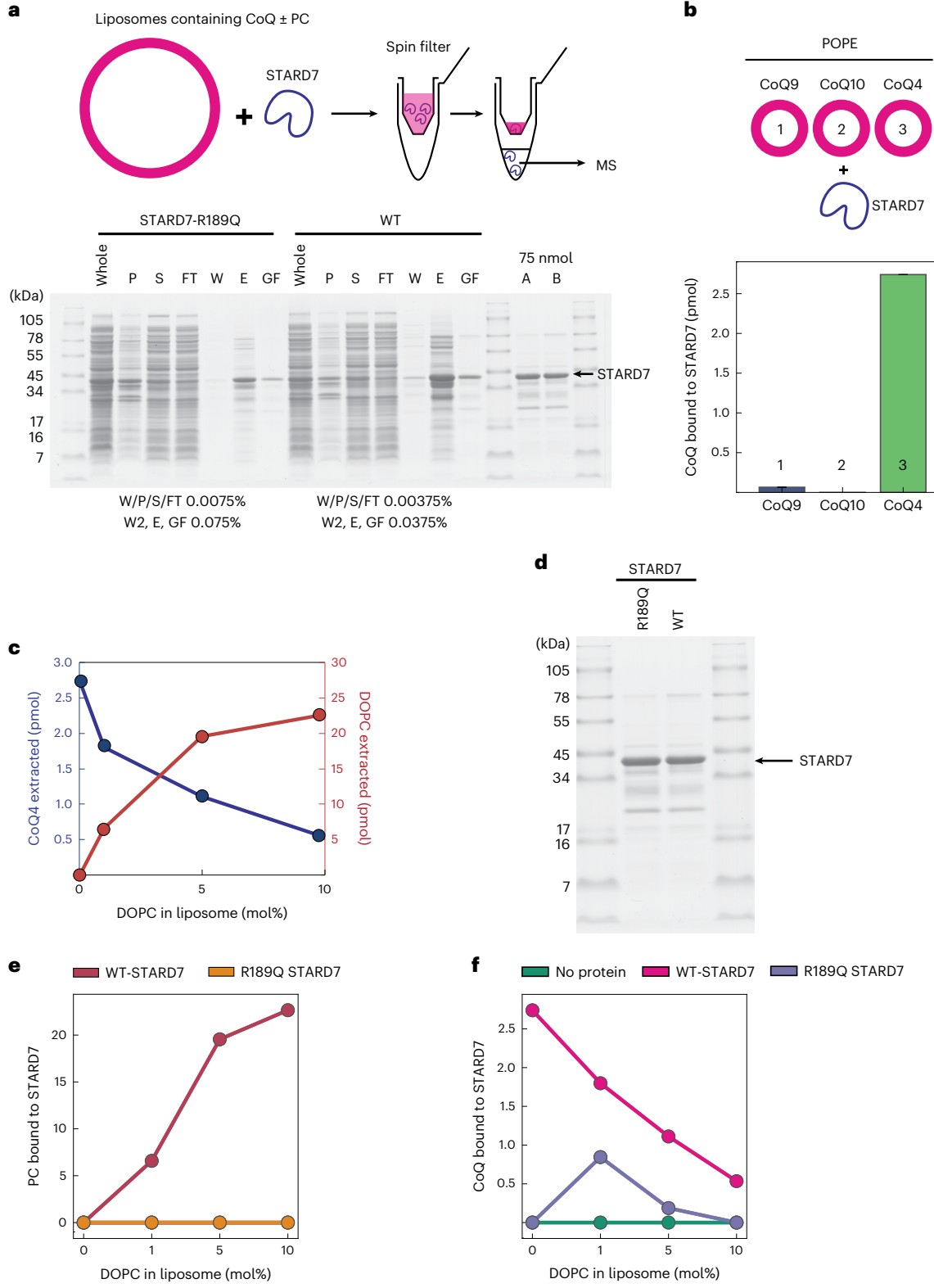

**Fig. 5 | STARD7 can extract CoQ4 from liposomes in vitro. a**, Scheme showing in vitro experiments, where liposomes containing different chain lengths of CoQ in the presence or absence of DOPC were incubated with STARD7 purified from *E. coli*. Top: after passing through a spin filter, lysate was analysed by MS. Purification of STARD7-his and its mutant variant (R189Q) is shown. C-terminally hexahistidine-tagged mature form of STARD7 and its mutant variant were expressed in T7 express *E. coli* cells. After mechanical lysis, the lysate was subjected to Ni-NTA affinity purification followed by gel filtration. P, pellet after lysis; S, supernatant after lysis; FT, flow-through fraction; W, wash fraction; E,

peak fraction eluted from HisTrap column; GF, peak fraction after gel filtration in HighLoad Superdex 200 pg column. Bottom: 75 nmol of each final sample was checked. **b**, CoQ extracted by STARD7 from liposomes containing either CoQ9 (1) or CoQ10 (2) or CoQ4 (3) in the absence of DOPC was measured by MS. Background signals in the absence of protein were subtracted. **c**, CoQ4 (pMol) and PC (pMol) extracted by STARD7. **d**, STARD7 and STARD7[R189Q] proteins purified from *E. coli*. **e,f**, PC (**e**) and CoQ (**f**) extracted by STARD7 or STARD7[R189Q] from liposomes containing increasing concentration of DOPC. Source numerical data are available in source data.

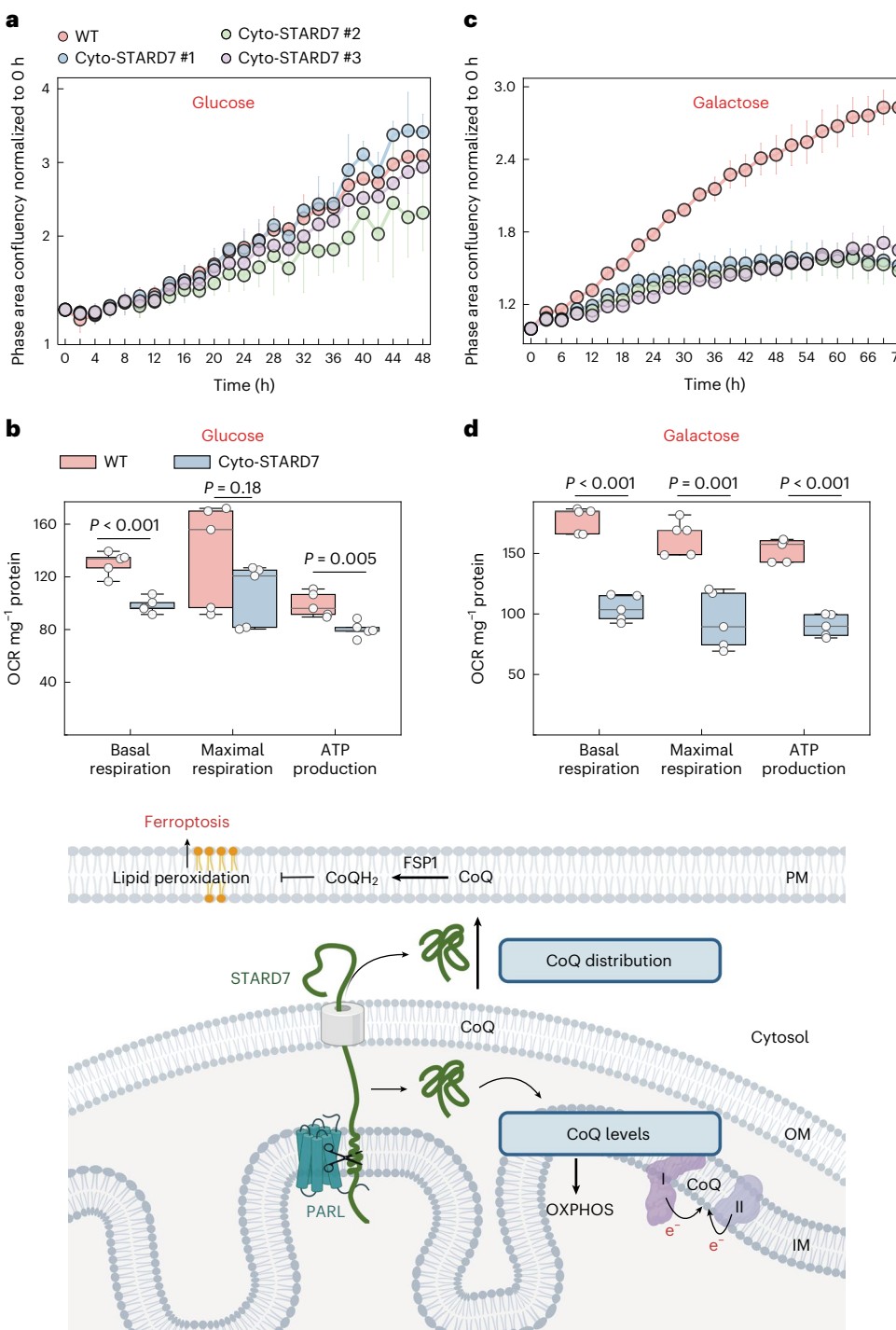

**Fig. 6 | Overexpression of cyto-STARD7 impairs cell growth and mitochondrial oxygen consumption under respiring conditions. a–d,** Growth rate (**a** and **c**) and OCR (**b** and **d**) of WT and cyto-STARD7-overexpressing cells in the presence of glucose (25 mM) (**a** and **c**) or galactose (10 mM) (**b** and **d**). In **a–d**, data are represented by 95% confidence interval of the mean and the *P* values were calculated using a two-tailed Student's *t*-test for unpaired comparisons where indicated. *n* = 5 (**a**, **b** and **d**) and *n* = 4 (**c**) biologically independent experiments. In **b** and **d**, the central band of each box is the 50% quantile, and the box defines the 25% (lower) and 75% (higher) quantile. The whiskers represent the minimum and maximum value in the data. Model illustrating the role of mito-STARD7 and cyto-STARD7 in CoQ biosynthesis and distribution, respectively. OM, mitochondrial outer membrane. Source numerical data are available in source data (image created with BioRender.com).

the cytosol, which allows coordination of CoQ synthesis and cellular CoQ distribution and balances mitochondrial respiration with the cellular defence against lipid peroxidation and ferroptosis.

PC transport across the IMS by mitochondrial STARD7 maintains CoQ levels independent of cytosolic STARD7, consistent with the critical role of mitochondria-localized STARD7 for cristae morphogenesis and respiration[40]. Mature STARD7 in the IMS is also sufficient to preserve the respiratory competence of *PARL*−/− cells, demonstrating that PARL regulates CoQ synthesis and respiration by processing of STARD7. As CoQ loss precedes OXPHOS deficiency in neuron-specific

*PARL*[−/−] mice[30], our results suggest that impaired STARD7 processing and CoQ synthesis contributes to respiratory complex III deficiencies and Leigh-like neurodegeneration in this model.

Whereas CoQ synthesis depends solely on mitochondrial STARD7, cytosolic STARD7 is required for CoQ transport from mitochondria to the PM and confers ferroptotic resistance to the cells. Interestingly, STARD7 is the second START-domain protein besides COQ10 (two orthologues exist in human) involved in CoQ metabolism. Yeast Coq10 is dispensable for CoQ synthesis but cells lacking Coq10 exhibit an increased sensitivity to oxidative stress[46]. Coq10 binds CoQ and is thought to chaperone CoQ to sites of functions within mitochondria[42,47,48]. Similarly, we observed direct CoQ4 binding to STARD7 in vitro, indicating that STARD7 directly affects CoQ transport in vivo. However, it remains to be determined whether STARD7 binding to CoQ results in a complete membrane extraction of CoQ, which appears unlikely considering the hydrophobicity of the polyprenoid tail of CoQ. Rather, a complex cellular machinery may drive the STARD7-dependent cellular distribution of CoQ, which mediates membrane remodelling and allows trafficking of newly synthesized CoQ, and perhaps additional membrane lipids, to other cellular membranes. As previous complementation studies in yeast indicated that exogenously added CoQ can be transported via the endocytic pathway[49,50], this may occur via vesicular transport. Moreover, it may involve contact sites between mitochondria and other cellular membranes, which often are characterized by lower PC concentrations and therefore represent membrane regions allowing CoQ binding by STARD7. Although further mechanistic studies are needed to establish how STARD7 affects CoQ transport, our results identify STARD7 as a cytosolic component that is required for and also limits CoQ transport from mitochondria to the PM, offering new possibilities to further unravel this intriguing cellular pathway.

STARD7 is targeted to mitochondria where it is cleaved by PARL upon import into mitochondria and partitions between the IMS and the cytosol[40]. Our results demonstrate that this allows coordination of CoQ synthesis and intracellular transport, adjusting bioenergetic CoQ functions in mitochondria with functions of CoQ as antioxidants in the PM. An imbalance in the cellular distribution of STARD7 has detrimental consequences for the cell: while low levels of STARD7 in the cytosol increase the susceptibility of the cells for ferroptosis, mitochondrial oxygen consumption and respiratory cell growth are impaired if levels of cytosolic STARD7 are increased relative to mitochondrial STARD7. Our findings thus reveal the need to balance synthesis and distribution of CoQ by PARL-mediated processing of STARD7. It is conceivable that two conserved atypical kinases of the UbiB family in the IMS, which were demonstrated to modulate CoQ export from yeast mitochondria, participate in this regulation[21].

While our previous results identified PARL as a pro-apoptotic protein[32], we demonstrate here that PARL and STARD7 maintain the CoQ- and FSP1-dependent antioxidant defence in the PM, limiting lipid peroxidation and ferroptosis. These findings highlight the important role of mitochondrial CoQ synthesis and cellular CoQ distribution for the suppression of ferroptosis, which occurs independent of the anti-ferroptotic function of GPX4. PARL and STARD7 may therefore represent promising targets to induce ferroptosis in tumours resistant to GPX4 inhibitors. On the other hand, ferroptosis has to be considered in genetic disorders caused by mutations in CoQ-synthesizing enzymes and in mitochondrial diseases with OXPHOS deficiencies[51]. As the loss of CoQ is emerging as a general consequence of OXPHOS defects[16,52], an increased susceptibility to ferroptosis may be of broad importance for the pathophysiology of these diseases.

## Online content

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

## Methods

The research conducted here complies with all the relevant ethical regulations. All animal experiments were approved by Landesamt für Natur, Umwelt und Verbraucherschutz Nordrhein-Westfalen, Germany and the Cologne Excellence Cluster on Cellular Stress Responses in Aging-Associated Diseases (CECAD) mouse facility regulations.

### Cell culture and generation of cell lines

HeLa (CCL-2) cells were purchased from American Type Culture Collection (ATCC). $PARL^{-/-}$, $STARD7^{-/-}$ and $SMAC^{-/-}$ HeLa cells have been described previously[32,40]. Clustered regularly interspaced short palindromic repeats (CRISPR)/Cas9 genome editing was performed to generate $PGAM5^{-/-}$ and $CLPB^{-/-}$ cells as described earlier[40]. The guide RNAs used for targeting PGAM5 deletion were 5′-GGCACATCTTCCTCATC-3′ and 5′-GTGGCCTTGGCTTTGTAG-3′ and those for CLPB deletion were 5′-GGCCGGAATGTGACTAC-3′ and 5′-GCCATGGCCCCGGAGCGT-3′. The human carcinoma colorectal HCT116 cell lines were bought from ATCC (catalogue number HeL). $PARL^{-/-}$ and $STARD7^{-/-}$ HCT116 cells were generated using plasmid pSpCas9 BB-2A-GFP (PX458, GenScript) containing guide RNA GAACCTCGAAGATCAGACCC for PARL and guide RNA ATCCAACTAACACAGTAGCG for STARD7. Cells were maintained in Dulbecco's modified Eagle's medium (DMEM) supplemented with GlutaMAX (Life Technologies) and 10% foetal bovine serum at 37 °C in the presence of 5% $CO_2$ and were routinely tested for *Mycoplasma* infections. For lentiviral transfection, HEK293T cells were transfected with pLVX-puro containing either mito-STARD7 or cyto-STARD7 (ref. [40]) for 24 h using Lenti-X Packaging Single Shots (Takara), after which medium was replaced with fresh DMEM. After 48 h of transfection the medium was collected and centrifuged at 1,000*g*. The virus-containing supernatant with 4 μg ml$^{-1}$ polybrene was used to infect 50% confluent $PARL^{-/-}$, $STARD7^{-/-}$ and WT cells for 24 h. After 2 days of transfection, cells were selected with puromycin 5 μg ml$^{-1}$.

### CoQ extraction and measurement

CoQ was extracted from cells using an extraction buffer (methyl tertiary-butyl ether, MeOH and $H_2O$ 50:30:20 (v:v:v)). The extraction buffer was prepared fresh for each extraction and was cooled down at −20 °C before use. A total of 300,000 cells per well were plated in a six-well plate, and CoQ was extracted the next day. For tracing experiments, cells were incubated with $^{13}C_6$-glucose for the indicated timepoints. Cells were trypsinized and centrifuged at 800*g* for 3 min and washed with phosphate-buffered saline. One millilitre pre-cooled extraction buffer was added to each sample, and samples were incubated at 1,500 rpm in thermomixer for 30 min at 4 °C. Samples were then centrifuged at 21,000*g* for 10 min. The pellets were used to determine protein concentrations using a bicinchoninic acid assay (BCA). The supernatant was mixed with 200 μl of pure MTBE and 150 μl of $H_2O$ (liquid chromatography (LC)–MS grade) to the cleared supernatant, and the mixture was incubated on the thermomixer at 15 °C for additional 10 min. Samples were centrifuged for 10 min at 15 °C and 16,000*g* to obtain phase separation. The lipid phase (top phase) was transferred in a new tube and dried in a Speed Vac concentrator at 20 °C at 1,000 rpm.

To determine CoQ levels, lipid pellets were re-suspended in 75 μl of ultrahigh-performance liquid chromatography (UPLC)-grade acetonitrile:isopropanol (70:30 (v:v)). Samples were vortexed for 10 s and incubated for 10 min on a thermomixer at 4 °C. Re-suspended samples were centrifuged for 5 min at 10,000*g* and 4 °C, before transferring the cleared supernatant to 2 ml glass vials with 200 μl glass inserts (Chromatographie Zubehör Trott). All samples were placed in an Acquity iClass UPLC (Waters) sample manager held at 6 °C. The UPLC was connected to a Tribrid Orbitrap HRMS, equipped with a heated ESI (HESI) source (ID-X, Thermo Fisher Scientific).

Of each lipid sample 2 μl was injected onto a 100 × 2.1 mm BEH C8 UPLC column, packed with 1.7 μm particles (Waters). The flow rate of the UPLC was set to 400 μl min$^{-1}$ and the buffer system consisted of buffer A (10 mM ammonium acetate and 0.1% acetic acid in UPLC-grade water) and buffer B (10 mM ammonium acetate and 0.1% acetic acid in UPLC-grade acetonitrile/isopropanol 7:3 (v/v)). The UPLC gradient was as follows: 0–1 min 45% A, 1–4 min 45–25% A, 4–12 min 25–11% A, 12–15 min 11–1% A, 15–20 min 1% A, 20–20.1 min 1–45% A and 20.1–24 min re-equilibrating at 45% A. This leads to a total runtime of 24 min per sample.

The ID-X mass spectrometer was operating in positive ionization mode scanning a mass range between *m*/*z* 150 and 1,500. The resolution was set to 120,000, leading to approximately four scans per second. The RF lens was set to 50%, while the AGC target was set to 40%. The maximal ion time was set to 100 ms, and the HESI source was operating with a spray voltage of 3.5 kV in positive ionization mode. The ion tube transfer capillary temperature was 300 °C, the sheath gas flow 60 arbitrary units (AU), the auxiliary gas flow 20 AU and the sweep gas flow was set to 1 AU at 340 °C.

All samples were analysed in a randomized run order. Targeted data analysis was performed using the quan module of the Trace-Finder 4.1 software (Thermo Fisher Scientific) in combination with a sample-specific compound database, derived from measurements of commercial reference compounds (Sigma).

### Data mining from CTRP

For generating the graph of Fig. 2a, the data were downloaded from previous study[53]. v21.data.gex_avg_log$_2$.txt and v21.data.auc_sensitivities.txt files were used, and all cell lines with growth-protocol 'adherent' were selected for each gene of interest. For all available compounds, table containing v21.data.gex_avg_log$_2$.txt and v21.data.auc_sensitivities.txt were merged on the master_ccl_id column. The Pearson's correlation was calculated between columns area_under_curve and mrna_expression_avg_log$_2$. Once all correlations between one gene and all compounds were calculated, *z*-scores were calculated as (correlation − mean (correlation))/standard deviation (correlation). This was done separately for all the genes of interest. The graph showing Pearson's correlation *z*-score for all the indicated genes was generated with Instant Clue software.

### OCR measurement

Oxygen consumption rate (OCR) was measured using Seahorse XF Analyzer. A total of 32,000 cells per well were plated in XF 96 cell culture microplates in DMEM medium supplemented with 10% FBS. The next day, DMEM was replaced by Seahorse XF medium supplemented with pyruvate (1 mM), glutamine (1 mM) and either glucose or galactose (10 mM). The OCR was measured according to the manufacturer's protocol (XF Cell Mito Stress Test Kit) at basal level and after injections with oligomycin (2 μM), FCCP (0.5 μM) and a combination of antimycin A (0.5 μM) and rotenone (0.5 μM). To normalize OCR data, protein concentration per well was determined using bicinchoninic acid assay. The data were analysed by Seahorse Wave Desktop software.

### Cell death assays

We used an Incucyte Live-Cell Analysis system (Sartorius) to monitor cell death. A total of 5,000 cells per well were plated in a 96-well flat-bottom plate. On the next day, medium was replaced with fresh medium and cells were treated with the indicated compounds in the presence of 150 nM Sytox green nucleic acid stain (Thermo Fisher Scientific, S7020). For live-cell analysis, images were captured every 3 h with the phase-contrast channel and green channel. To analyse the dead cell area, a mask was created by using the Incucyte base analysis software with basic analyser for both phase and green channels. Area occupied by green objects was normalized to phase area per well to measure cell death. All compounds were bought from Sigma, including erastin (E7781), RSL3 (SML2234), ferrostatin-1 (SML 0583), CoQ2 (C8081), QVD (SML 0063), FSP1 inhibitor (SML2749), 4-CBA (135585) and arachidonic acid 20:4 (A9673), which was used as PUFA.

## Isolation of mitochondrial and plasma membrane fractions

Mitochondrial and plasma membrane fractions were isolated from cells using differential centrifugation as described previously[32]. Briefly, cells were homogenized in isolation buffer (20 mM HEPES/KOH pH 7.4, 220 mM mannitol, 70 mM sucrose and 1 mM EGTA) using a potter at 1,000 rpm. Whole cell (WC) sample was collected after homogenization. Homogenized cells were centrifuged at 8,000$g$ for 10 min at 4 °C to isolate mitochondria. The supernatant was centrifuged again at 8,000$g$ for 10 min at 4 °C to collect further mitochondrial fraction. Then, 8,000$g$ pellets were pooled together to collect mitochondrial fraction (fraction 1). To remove remaining mitochondrial contaminations, the supernatant was centrifuged at 12,000$g$ for 15 min at 4 °C. To isolate the PM fraction, the supernatant was centrifuged at 40,000$g$ for 45 min at 4 °C and then again at 100,000$g$ for 1 h at 4 °C. WC sample, 8,000$g$ (fraction 1), 12,000$g$ (fraction 2), 40,000$g$ (fraction 3) and 100,000$g$ (fraction 4) were used for proteomics analysis, and 8,000$g$ and 40,000$g$ pellets were used to extract CoQ.

## Expression and purification of STARD7

Gene encoding hexahistidine-tagged version of mature STARD7 (76–370) was amplified from human complementary DNA and cloned into pET16b. R189Q variant was generated by site-directed mutagenesis PCR. T7 Express (NEB) carrying the construct was cultivated in LB medium containing 100 µg ml⁻¹ ampicillin. After incubation at 37 °C for 3 h (OD$_{600}$ ~0.3), the culture was shifted to 18 °C for 1 h, and then STARD7 was expressed by adding IPTG (0.3 mM) for 14 h. *E. coli* cells were lysed in lysis buffer (20 mM Tris–HCl, pH 8, 250 mM NaCl, 1× complete protease inhibitor mix (Roche), 20 mM imidazole, 100 U ml⁻¹ DNase I, 1 mM MgCl$_2$ and 10 mM tris(2-carboxyethyl)phosphine (TCEP), 25 ml per gram wet cell weight) by Emusiflex C-5 (Avestin). The lysate was spun at 30,000$g$ for 20 min, and the supernatant was filtrated through a 0.45 µm vacuum filter. The lysate was applied on HisTrap column (Cytiva). After washing the column by buffer C (10 mM Tris–HCl pH 8, 250 mM NaCl and 1 mM dithiothreitol) containing 40 mM imidazole, proteins were eluted from column by buffer C containing 250 mM imidazole. The peak elution fractions were pooled and subjected to size exclusion chromatography using HiLoad superdex 200 pg column (GE Healthcare) in buffer C. 7-nitrobenz-2-oxa-1,3-diazol-4-yl (NBD)-PC transfer activities were determined in elution fractions, and fractions containing the highest PC transfer activity were combined. Protein concentrations in samples were determined by absorbance at 280 nm and extinction coefficient of the protein. Samples were divided into small aliquots and then stored at −80 °C.

## Lipid extraction assay

All phospholipids were obtained from Avanti Polar Lipids. Phospholipids and CoQ in stock solutions in chloroform or methanol were mixed at the desired molar ratio (71% 1-palmitoyl-2-oleoyl-sn-glycero-3-phosphoethanolamine (POPE), 0.3% tetraoleoyl cardiolipin (TOCL), 2.1% 1-palmitoyl-2-oleoyl-sn-glycero-3-phosphatidylserine (POPS), 0.4% dioleoylphosphatidic acid (DOPA), 0.2% dioleoylphosphatidylglycerol (DOPG), 13.4% egg phosphatidylinositol (PI), 2% 17:0 phosphatidylethanolamine (PE), 0.6 % rhodamine PE and 10% of either CoQ or dioleoylphosphatidylcholine (DOPC)), and the solvent was evaporated under a flow of nitrogen. The lipid film was hydrated at 1 mM total lipid in assay buffer (20 mM Tris–HCl pH 8 and 150 mM NaCl). The suspension was incubated at room temperature for 1 h and extruded through polycarbonate filters of 0.1 µm pore size using a mini-extruder (Avanti Polar Lipids). In a standard lipid extraction assay, the reconstituted liposomes (final total lipid concentration is set to 100 µM) were incubated with 4 µM of purified STARD7 for 10 min at 37 °C in 250 µl final volume. After cooling down on ice, the mixture was filtered through Amicon ultra 0.5 filter unit (Millipore, 15 min, 14,000$g$, 4 °C) and the flow through was recovered. A portion of flow through was subjected to SDS–PAGE to assess protein recovery. The rest (150 µl, 60% of total)

was subjected to MS. No contamination of liposomes in flow through was certified by rhodamine fluorescence of rhodamine-PE and the signal of POPE/17:0 PE on MS analysis.

## Protein digestion for proteomics experiments

Forty microlitres of 4% SDS in 100 mM HEPES/KOH pH 8.5 was pre-heated to 70 °C and added to the cell pellet for further incubation for 10 min at 70 °C on a ThermoMixer (shaking 550 rpm). The protein concentration was determined using the 660 nm Protein Assay (Thermo Fisher Scientific, #22660). Fifty micrograms of protein was subjected to tryptic digestion. Proteins were reduced (10 mM TCEP) and alkylated (20 mM chloroacetamide (CAA)) in the dark for 45 min at 45 °C. Samples were subjected to SP3-based digestion 1. Washed SP3 beads (SP3 beads (Sera-Mag magnetic carboxylate modified particles (hydrophobic, GE44152105050250), Sera-Mag magnetic carboxylate modified particles (hydrophilic, GE24152105050250) from Sigma Aldrich) were mixed equally, and 3 µl of bead slurry was added to each sample. Acetonitrile was added to a final concentration of 50% and washed twice using 70% ethanol ($V$ (volume) = 200 µl) on an in-house-made magnet. After an additional acetonitrile wash ($V$ = 200 µl), 5 µl digestion solution (10 mM HEPES/KOH pH 8.5 containing 0.5 µg trypsin (Sigma) and 0.5 µg LysC (Wako)) was added to each sample and incubated overnight at 37 °C. Peptides were desalted on a magnet using 2× 200 µl acetonitrile. Peptides were eluted in 10 µl 5% dimethyl sulfoxide (DMSO) in LC–MS water (Sigma Aldrich) in an ultrasonic bath for 10 min. Formic acid and acetonitrile were added to a final concentration of 2.5% and 2%, respectively. Samples were stored at −20 °C until subjected to LC–MS/MS analysis. Before measurement, samples were re-suspended in 10 µl of 2% acetonitrile and 2% formic acid containing internal standard index retention time (iRT) peptides.

## LC and MS

Instrumentation consisted out of an nLC1200 LC system coupled via a nano-electrospray ionization source to a quadrupole-based mass spectrometer (Exploris 480, Thermo Fisher Scientific). For peptide separation an in-house packed column (inner diameter 75 µm, length 40 cm) was used. A binary buffer system (A: 0.1% formic acid and B: 0.1% formic acid in 80% acetonitrile) was applied as follows: Linear increase of buffer B from 4% to 27% within 69 min, followed by a linear increase to 45% within 5 min. The buffer B content was further ramped to 65% within 5 min and then to 95% within 6 min, and 95% buffer B was kept for further 10 min to wash the column. Before each sample, the column was washed using 5 µl buffer A and the sample was loaded using 8 µl buffer A. For MS spectra acquisition, the RF lens amplitude was set to 55%, the capillary temperature was 275 °C and the polarity was set to positive. MS1 profile spectra were acquired using a resolution of 120,000 (spatial proteomics) or 60,000 (brain whole proteome) (at 200 $m/z$) at a mass range of 320–1,150 $m/z$ and an AGC target of $1 \times 10^6$. For MS/MS independent spectra acquisition, 48 windows were acquired at an isolation $m/z$ range of 15–17 Th and the isolation windows overlapped by 1 Th. The fixed first mass was set to 200. The isolation centre range covered a mass range of 350–1,065 $m/z$. Fragmentation spectra were acquired at a resolution of 15,000 at 200 $m/z$ using a maximal injection time of 23 ms and stepped normalized collision energies of 26, 28 and 30. The default charge state was set to 3. The AGC target was set to 900%. MS2 spectra were acquired as centroid spectra.

## Mass spectrometric data analysis–brain whole proteome

The software DIA-NN (v1.8) (ref. [53]) was used to analyse data independent raw files. The spectral library was created using the reviewed only Uniport *Mus musculus* (downloaded 06.2019) reference proteome with the 'Deep learning-based spectra and RTs prediction' turned on. Protease was set to trypsin, and a maximum of one miscleavage was allowed. N-term M excision was set as a variable modification and carbamidomethylation at cysteine residues was set as a fixed modification.

The peptide length was set to 7–30 amino acids, and the precursor $m/z$ range was defined from 340 to 1,200 $m/z$. The option 'Quantitative matrices' was enabled.

The false discovery rate (FDR) was set to 1%, and the mass accuracy (MS2 and MS1) as well as the scan window was set to 0 (automatic inference via DIA-NN). Match between runs (MBR) was enabled. The neuronal network classifier worked in 'double pass mode', and protein interference was set to 'Isoform IDs'. The quantification strategy was set to 'robust LC (high accuracy)', and cross-run normalization was defined as 'RT-dependent'. The 'pg' (protein group) output (MaxLFQ intensities2) was further processed using Instant Clue[54,55]. Statistical significance was assessed using a two-sided $t$-test on $\log_2$-transformed label-free quantitation (LFQ) intensities. A permutation-based approach was used to control the FDR to 0.05.

## Mass spectrometric data analysis—spatial proteomics

Data were analysed using Spectronaut4 (15.7.220308.50606) and the human Uniprot reference proteome (downloaded 06.2019) using the implemented directDIA mode. Minimal peptide length was set to 7, and the maximum number of miscleavages was 2. Dynamic mode to estimate mass tolerances for MS1 and MS2 was used. The maximum number of variable modifications was set to 5. Acetylation at the protein N-term and methionine residues oxidization were set as a variable modification. Carbamidomethylation at cysteine residues was defined as a fixed modification. The iRT–rt regression type was set to local (non-linear) regression. Peptide grouping from precursors was done using the stripped peptide sequences. Imputing was set off, and cross-run normalization was enabled. Protein interference was performed using the implemented 'IDPicker' algorithm. The peptide, peptide-spectrum match and protein group FDR were set to 0.05. The intensity-based absolute quantification (iBAQ) intensity was calculated and is reported. -P indicates pellet fraction and -SN the supernatant fraction after centrifugation. For visualization, iBAQ intensities were scaled between 0 and 1 and visualized using hierarchical clustering (Euclidean distance, complete method) using the Instant Clue software. The MitoCarta 3.0 was used to define mitochondrial proteins. The set of PM proteins was extracted by using the Uniprot Gene Ontology cellular compartment information. Proteins that are annotated with 'plasma membrane (GO:0005886)' but not with nucleus (GO:0005634), cytosol (GO:0005829), cytoplasm (GO:0005737), endoplasmic reticulum (GO:0005783) or Golgi apparatus (GO:0005794) and that are also not part of the MitoCarta 3.0 repository were considered as the PM proteins. Supplementary Table 2 contains detailed information about the annotation process. Heat maps were created using the Instant Clue software using the complete method and correlation metrics to cluster rows. Columns were not clustered.

## Mice tissue collection

CRISPR/Cas9 system was used to generate $PARL^{-/-}$ C57BL/6N mice using guide RNA targeting exon 2 5′-CCCTTCCCCCTATCCTATAAGAA-3′ and exon 5 5′-GGACAGCATACGGCCACAAAAGG-3′. Mice were maintained at the specific-pathogen-free animal facility of the CECAD Research Centre with 12 h light cycle and regular chow diet. To perform proteomics and metabolomics analysis on brain and/or heart tissue, 5-week-old male WT and $PARL^{-/-}$ mice were killed by cervical dislocation. Brains and hearts were extracted from mice and immediately frozen in the liquid nitrogen. Tissues were then homogenized using mortar and pestle on dry ice and used for CoQ extraction. The protein pellets from CoQ extraction were further used for proteome analysis.

## Statistics and reproducibility

All the independent experiments or biological samples are represented in the graphs. Instant Clue software 3.0 was used to analyse all the datasets[56]. Data are represented by 95% confidence interval of the mean to show statistically significant differences between the groups.

To compare two groups, $P < 0.05$ was considered significant and a two-tailed Student's $t$-test was performed for unpaired comparisons. No statistical method was used to pre-determine sample size. No data were excluded from the analyses. The investigators for proteomics and metabolomics measurement were blinded to allocation during experiments and samples were randomized. The investigators were not blinded for all other experiments, and samples were not randomized.

## Antibodies

The following commercial antibodies were used: SMAC (MBL, JM-3298-100) dilution 1:1,000, STARD7 (Proteintech 15689-1-AP) dilution 1:2,500, FLAG (WAKO 018-22381) dilution 1:1,000, SDHA (Abcam Ab14715) dilution 1:10,000, He) dilution 1:1,000, CLPP (Sigma HPA010649) dilution 1:1,000, YME1L (Proteintech 11510-1-AP) dilution 1:1,000, MIC60 (Nobus Biologicals 100-1919) dilution 1:1,000, VDAC2 (Proteintech 11663-1-AP) dilution 1:1,000. PARL antibody is defined previously[32], dilution 1:1,000. All antibodies were diltuted in 5% milk-TBST, and nitrocellulose membranes were incubated overnight at 4 °C.

## Reporting summary

Further information on research design is available in the Nature Portfolio Reporting Summary linked to this article.

## Data availability

MS data have been deposited in ProteomeXchange with the primary accession code PXD033064. Previously published gene expression data that were re-analysed here are available through the Cancer Therapeutics Response Portal (http://www.broadinstitute.org/ctrp)[3]. Source data are provided with this paper. All other data supporting the findings of this study are available from the corresponding author on reasonable request.

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

## Acknowledgements

We thank F. Metge (Bioinformatics Facility, MPI) for help with the data mining from CTRP, C. Münch for providing $PINK1^{-/-}$ cells, the CECAD mouse facility, H.-G. Sprenger and A. Hesseling for help with generating $PARL^{-/-}$ mice, D. Diehl for technical help for proteomics studies, S. von Karstedt (CECAD) for help in setting up ferroptosis experiments and Y. Hinze (Metabolomics Facility, MPI Biology of Ageing) for providing technical help with CoQ measurements. This work was supported by Alexander von Humboldt-Foundation (Ref ITA 1201655 HFST-P) to S.D. and grants from the Deutsche Forschungsgemeinschaft (DFG, German Research Foundation) SFB1403 Projektnummer 414786233 and the German-Israeli-Project (DIP, RA1028/10-2) to T.L.

## Author contributions

S.D. and T.L. designed the project. S.D. performed and analysed most of the experiments. M.O. generated RQ mutant cell lines. T.B. and S.D. performed and analysed fractionation experiments. T.T. performed in vitro experiments. H.N. measured proteomics, and P.G. measured

lipids. K.R. helped in setting up ferroptosis assays. K.L. helped technically. S.D. and T.L. prepared manuscript. All authors discussed the results and contributed to the manuscript.

## Funding

## Competing interests
The authors declare no competing interests.

## Additional information
**Extended data** is available for this paper at

**Supplementary information** The online version
contains supplementary material available at

**Correspondence and requests for materials** should be addressed to
Thomas Langer.

**Peer review information** *Nature Cell Biology* thanks Michael Murphy
and the other, anonymous, reviewer(s) for their contribution to the
peer review of this work. Peer reviewer reports are available.

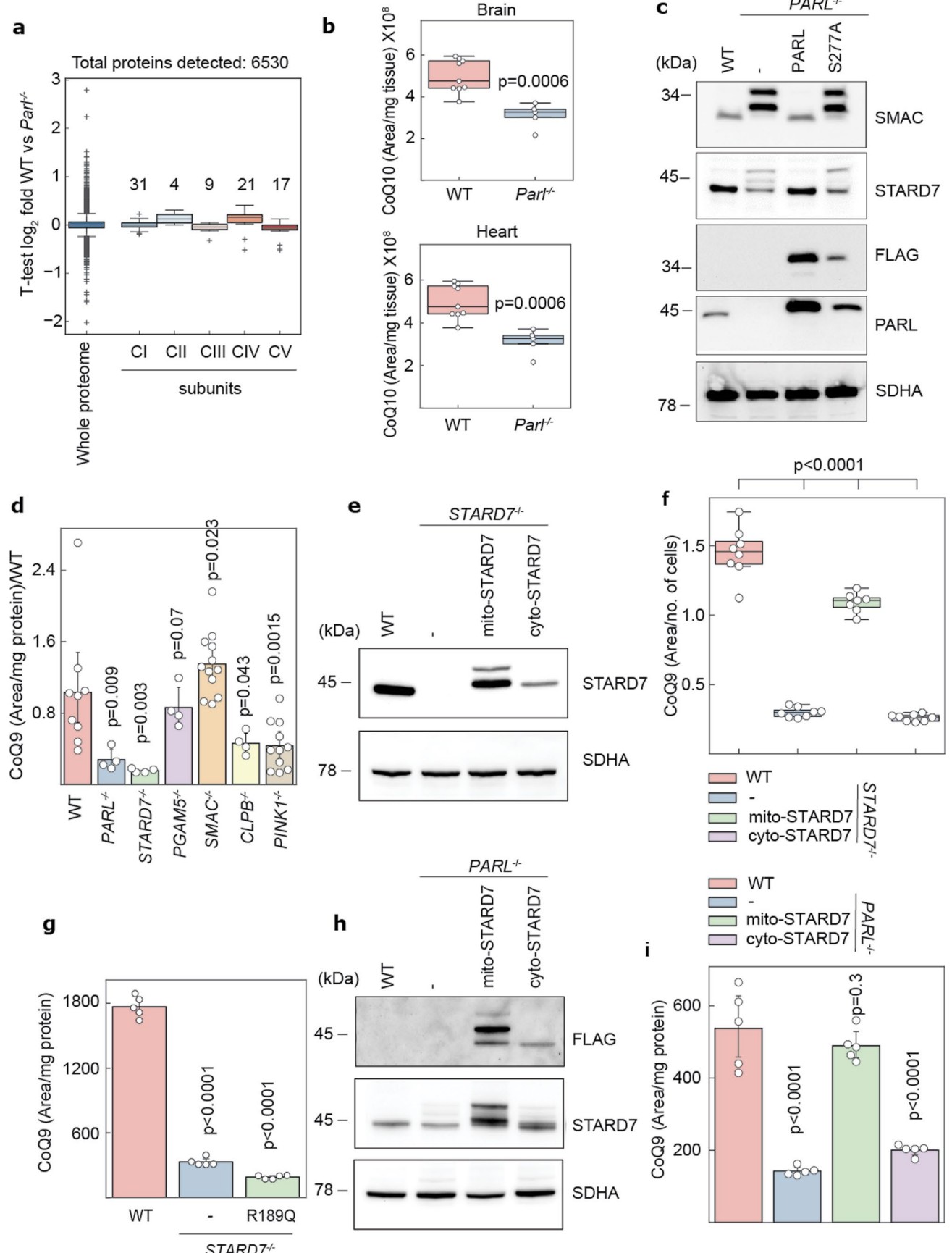

**Extended Data Fig. 1 | See next page for caption.**

**Extended Data Fig. 1 | PARL mediated STARD7 cleavage is required for maintaining CoQ levels. a**, Quantification of subunits of OXPHOS complexes I, II, III and IV and of the $F_1F_O$ ATP synthase (complex V), detected by mass spectrometry of 5-week-old male wild-type and $PARL^{-/-}$ brain tissues. Numbers of subunits detected for each complex are noted in the graph (n = 5 male mice). **b**, CoQ9 levels measured by mass spectrometry in brain and heart of 5-week-old male WT and $PARL^{-/-}$ mice (n = 9 WT and n = 5 $PARL^{-/-}$ animals). **c**, Immunoblot analysis of WT cells, $PARL^{-/-}$ cells and $PARL^{-/-}$ cells expressing PARL or PARL$^{S277A}$. **d**, CoQ9 abundance in HeLa cells lacking the indicated PARL substrate proteins (n = 10 WT, n = 4 $STARD7^{-/-}$, $PARL^{-/-}$, $PGAM5^{-/-}$, $CLPB^{-/-}$, n = 12 $SMAC^{-/-}$, n = 11 $PINK1^{-/-}$ biologically independent samples). **e**, Immunoblot analysis of $STARD7^{-/-}$ cells expressing mito-STARD7 or cyto-STARD7. **f,g**, CoQ9 levels measured in WT cells, $STARD7^{-/-}$ cells, and in $STARD7^{-/-}$ cells complemented with mito-STARD7 or cyto-STARD7 (n = 8 WT, $STARD7^{-/-}$ and $STARD7^{-/-}$ complemented with cyto-STARD7, n = 7 $STARD7^{-/-}$ complemented with mito-STARD7 biologically independent samples) (**f**) or mito-STARD7$^{R189Q}$ (**g**). **h**, Immunoblot analysis of $PARL^{-/-}$ cells expressing mito-STARD7 or cyto-STARD7. **i**, CoQ9 abundance in WT cells, $PARL^{-/-}$ cells and $PARL^{-/-}$ cells expressing mito-STARD7 or cyto-STARD7. **g,i**, n = 5 biologically independent samples. **c,e,h**, Representative immunoblots of n = 2 independent experiments. **a–b,f**, The central band of each box is the 50% quantile, and the box defines the 25% (lower) and 75% (higher) quantile. The whiskers represent the minimum and maximum value in the data and outliers are indicated by a + sign (greater distance than 1.8 times inter quantile range away from the median). **b,d,f,g,i**, Data is represented by 95% confidence interval of the mean and the P values were calculated using a two-tailed Student's $t$-test for unpaired comparisons where indicated.

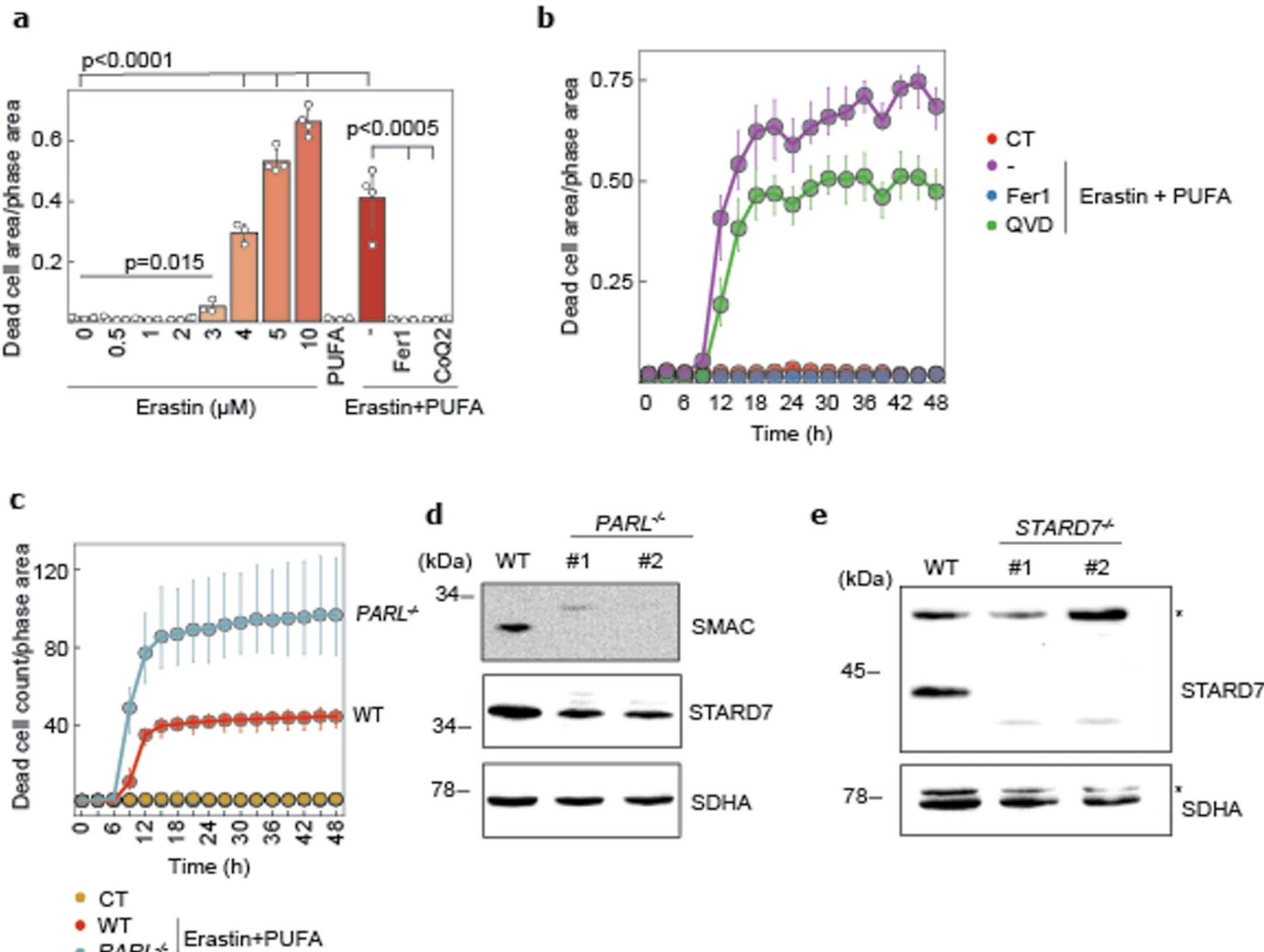

**Extended Data Fig. 2 | PARL and STARD7 protect cells against ferroptosis.**
**a,b**, |Cell death measurement after 24 h (**a**) or kinetics (**b**) in WT cells treated with indicated concentrations of erastin (**a**) or combination of erastin (3 μM) and PUFA (arachidonic acid 20:4, 40 μM) (**a,b**), in the presence of ferroptosis inhibitors Fer1 (1 μM) or CoQ2 (1 μM) and apoptosis inhibitor QVD (1 μM) as indicated. WT cells treated with DMSO were used as control (CT). **c**, Cell death kinetics in WT and *PARL⁻/⁻* cells treated with erastin (3 μM) and PUFA (40 μM). **d,e**, Immunoblot analysis of WT, *PARL⁻/⁻* (**d**) and *STARD7⁻/⁻* (**e**) human colorectal cancer HCT116 cells. #1 and #2 represent two different monoclonal cell lines of the indicated genotype. **a**, n = 4 and **b–c**, n = 3 biological independent experiments. **d–e**, Representative immunoblots of n = 2 independent experiments. **a–c**, Data is represented by 95% confidence interval of the mean and the *P* values (**a**) were calculated using a two-tailed Student's *t*-test for unpaired comparisons.

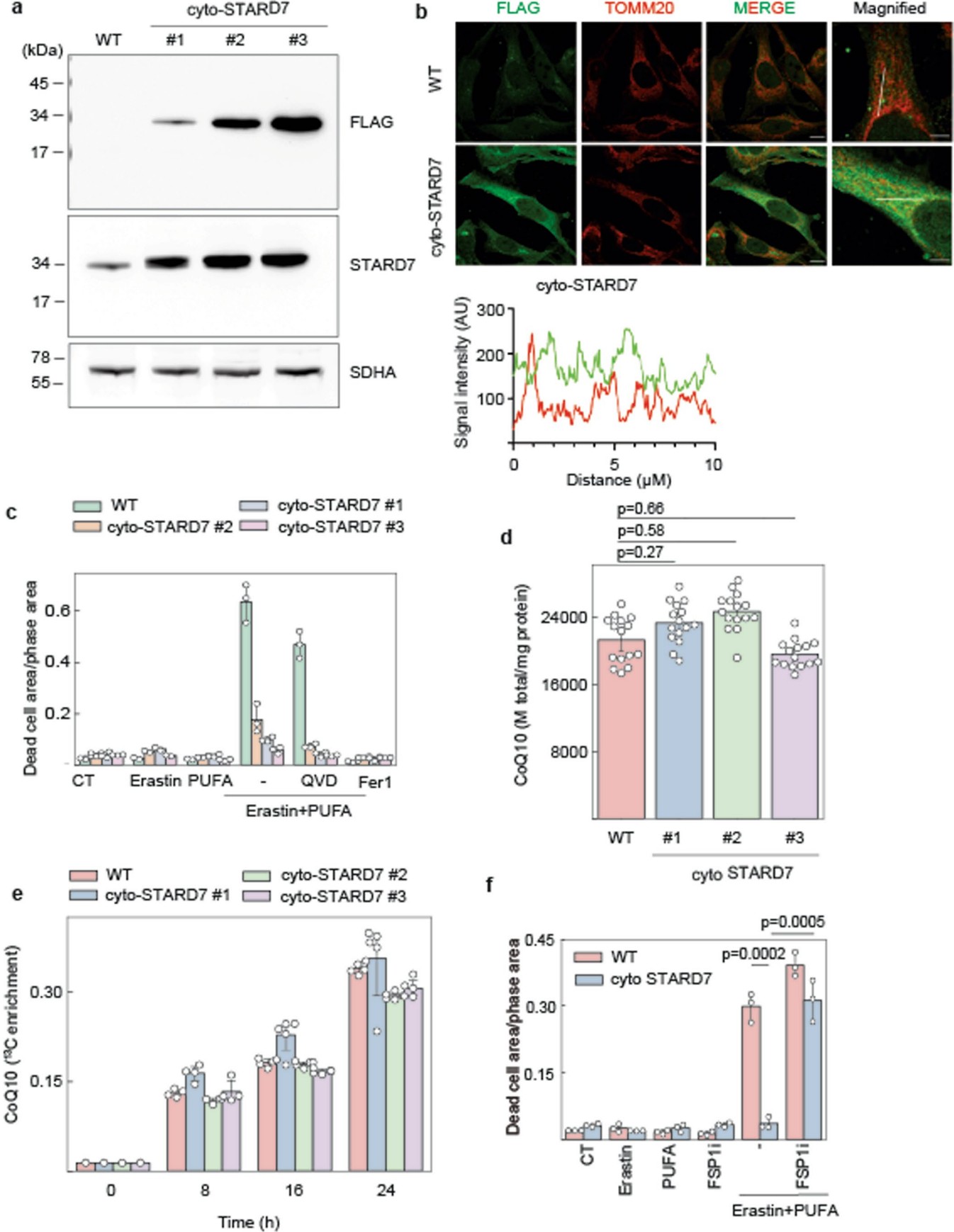

**Extended Data Fig. 3 | See next page for caption.**

**Extended Data Fig. 3 | Cytosolic STARD7 is a suppressor of ferroptosis.**
**a**, Immunoblot analysis of WT cells and WT cells overexpressing FLAG-tagged cyto-STARD7. #1, #2 and #3 represent different monoclonal cell lines selected from a polyclonal population. **b**, Localization of cyto-STARD7 in WT and cyto-STARD7 overexpressing cells was determined by immunofluorescence in HeLa cells. Cells are stained with antibodies against FLAG (green) to detect overexpressed STARD7 and TOMM20 (red) to detect mitochondria. Immunofluorescence images were taken via confocal microscopy with 1,000x magnification. A merge of both, as well as a magnified images are shown. Scale bars are 10 µm (magnified image). Graph shows the signal intensity plots of STARD7 (green) and TOMM20 (red) through line segment analysis (white line) in magnified image, n = 2 independent experiments. **c**, Cell death measured in WT and cyto-STARD7 overexpressing monoclonal cell lines treated either with erastin (3 µM) (n = 3) or PUFA (arachidonic acid 20:4, 40 µM) (n = 2), or combination of erastin and PUFA (n = 3) in the presence of the ferroptosis inhibitor Fer1 (1 µM) or the apoptosis inhibitor QVD (1 µM) as indicated (n = 3). **d**,**e**, Total CoQ levels (n = 15 independent biological samples) (**d**) and CoQ synthesis (**e**) measured in WT and cyto-STARD7 overexpressing cells incubated with $^{13}C_6$ glucose for the indicated time points. **f**, Ferroptosis was induced in WT and cyto-STARD7 overexpressing cells by either inhibiting GPX4 arm via erastin (1 µM) and PUFA (40 µM) or by inhibiting both GPX4 and CoQ-FSP1 arm via combination of erastin+PUFA and iFSP1 (1 µM). Cell death was measured after 24 h. **a**, n = 2 and **f**, n = 3 independent experiments and **e**, n = 5 (with one blank at 0 h from each genotype) biologically independent samples. **c**–**f**, Data is represented by 95% confidence interval of the mean and the P values were calculated using a two-tailed Student's t-test for unpaired comparisons where indicated.

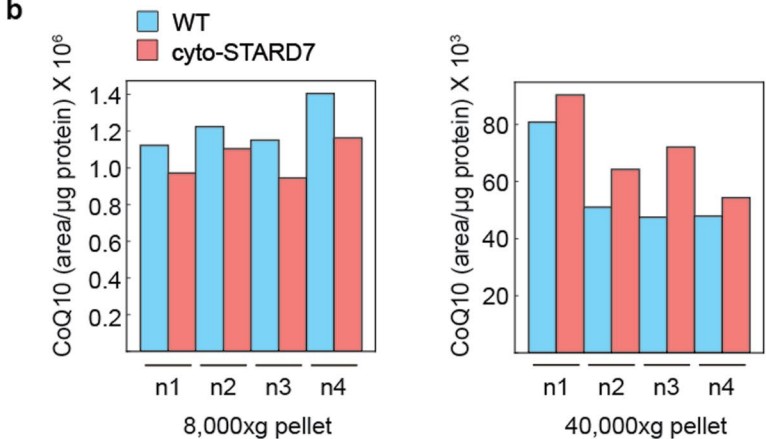

**Extended Data Fig. 4 | Cytosolic STARD7 is required for CoQ transport from the mitochondria to the plasma membrane. a**, Immunoblot analysis of cellular fractions obtained by differential centrifugation of lysates of WT and cyto-STARD7 overexpressing cells. The highlighted fractions 8 K and 40 K represent mitochondrial fraction and plasma membrane (PM) fractions, respectively, and were used to measure CoQ and other lipids. WC, whole cell; PNS, post-nuclear supernatant, n = 2 independent biological experiments. **b**, Raw data representation of 4 independent fractionation experiments from Fig. 4g showing CoQ10 levels measured by mass spectrometry in 8,000xg (8 K) and 40,000xg (40 K) pellet.

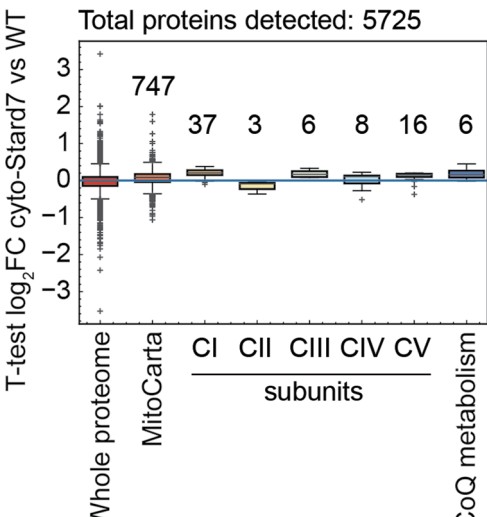

**Extended Data Fig. 5 | Overexpression of STARD7 limits respiratory cell growth. a**, Proteome analysis of all the detected OXPHOS subunits, including complex I, II, III, IV and V, and CoQ biosynthetic machinery in WT and cyto-STARD7 overexpressing cells. Number of subunits detected for each OXPHOS complex and CoQ biosynthetic machinery are noted in the graph. The central band of each box is the 50% quantile, and the box defines the 25% (lower) and 75% (higher) quantile. The whiskers represent the minimum and maximum value in the data and outliers are indicated by a + sign (greater distance than 1.8 times inter quantile range away from the median). Data represents the boxplots from n = 5 independent biological samples.

# Reporting Summary

## Statistics

For all statistical analyses, confirm that the following items are present in the figure legend, table legend, main text, or Methods section.

| n/a | Confirmed | |
|---|---|---|
| ☐ | ☒ | The exact sample size (*n*) for each experimental group/condition, given as a discrete number and unit of measurement |
| ☐ | ☒ | A statement on whether measurements were taken from distinct samples or whether the same sample was measured repeatedly |
| ☐ | ☒ | The statistical test(s) used AND whether they are one- or two-sided *Only common tests should be described solely by name; describe more complex techniques in the Methods section.* |
| ☒ | ☐ | A description of all covariates tested |
| ☒ | ☐ | A description of any assumptions or corrections, such as tests of normality and adjustment for multiple comparisons |
| ☐ | ☒ | A full description of the statistical parameters including central tendency (e.g. means) or other basic estimates (e.g. regression coefficient) AND variation (e.g. standard deviation) or associated estimates of uncertainty (e.g. confidence intervals) |
| ☒ | ☐ | For null hypothesis testing, the test statistic (e.g. *F*, *t*, *r*) with confidence intervals, effect sizes, degrees of freedom and *P* value noted *Give P values as exact values whenever suitable.* |
| ☒ | ☐ | For Bayesian analysis, information on the choice of priors and Markov chain Monte Carlo settings |
| ☒ | ☐ | For hierarchical and complex designs, identification of the appropriate level for tests and full reporting of outcomes |
| ☐ | ☒ | Estimates of effect sizes (e.g. Cohen's *d*, Pearson's *r*), indicating how they were calculated |

*Our web collection on statistics for biologists contains articles on many of the points above.*

## Software and code

Policy information about availability of computer code

| Data collection | No software was used |
|---|---|
| Data analysis | Data was analyzed using Instant clue software 0.11/0.53. Raw data (Mass spec): Spectraraut 15.7.220308.50606 (Fractionation data). DIA.NN 1.8 (tissue samples). https://github.com/hnolCol/instantclue. For metabolomics data (Lipids) Tracefinder 4.1/5.1 from Thermo Fischer was used. Incucyte Base Analysis Software (Included with Incucyte® Live-Cell Analysis System) was used for cell death analysis. Seahorse data was analyzed by Seahorse Wave Desktop Software 2.6. |

For manuscripts utilizing custom algorithms or software that are central to the research but not yet described in published literature, software must be made available to editors and reviewers. We strongly encourage code deposition in a community repository (e.g. GitHub). See the Nature Portfolio guidelines for submitting code & software for further information.

## Data

Policy information about availability of data

All manuscripts must include a data availability statement. This statement should provide the following information, where applicable:
- Accession codes, unique identifiers, or web links for publicly available datasets
- A description of any restrictions on data availability
- For clinical datasets or third party data, please ensure that the statement adheres to our policy

Mass spectrometry data have been deposited in ProteomeXchange with the primary accession code PXD033064. Previously published gene expression data that were re-analyzed here are available through the Cancer Thera-peutics Response Portal (http://www.broadinstitute.org/ctrp) [3]. Numerical source data and unprocessed blots have been provided in Source Data. All other data supporting the findings of this study are available from the corresponding author on

March 2021

reasonable request.

# Field-specific reporting

Please select the one below that is the best fit for your research. If you are not sure, read the appropriate sections before making your selection.

☒ Life sciences ☐ Behavioural & social sciences ☐ Ecological, evolutionary & environmental sciences

For a reference copy of the document with all sections, see nature.com/documents/nr-reporting-summary-flat.pdf

# Life sciences study design

All studies must disclose on these points even when the disclosure is negative.

| | |
|---|---|
| Sample size | No sample size was calculated. No statistical test was performed to determine the sample size. The sample size included at lease 3 biological replicates where statistical test was performed. |
| Data exclusions | No data was excluded |
| Replication | All n are shown separately in each figure. All attempts at replication were successful. |
| Randomization | Proteomics data and metabolomics data were randomised. For cell culture experiments, samples were allocated in different groups either based on genotypes or based on different treatments at different time-points. Whenever possible different experimental groups were handled together. While doing incucyte experiments, control/wildtype were measured first and then the treated/knockout samples. Covariates do not apply to cell culture experiments, since all the experiments were performed in the same cell culture room, all the cells were cultured in the same DMEM medium and put in the same incubators. |
| Blinding | Proteomics and metabolomics measurements were blinded. Samples were not blinded for tissue culture experiments. Since most of the experiments were done by the first author and samples were labeled in the tissue culture for the treatments/time-points, it was hard to not know the groups and therefore, the experimenter was always aware of the experimental groups. |

# Reporting for specific materials, systems and methods

We require information from authors about some types of materials, experimental systems and methods used in many studies. Here, indicate whether each material, system or method listed is relevant to your study. If you are not sure if a list item applies to your research, read the appropriate section before selecting a response.

### Materials & experimental systems

| n/a | Involved in the study |
|---|---|
| ☐ | ☒ Antibodies |
| ☐ | ☒ Eukaryotic cell lines |
| ☒ | ☐ Palaeontology and archaeology |
| ☐ | ☒ Animals and other organisms |
| ☒ | ☐ Human research participants |
| ☒ | ☐ Clinical data |
| ☒ | ☐ Dual use research of concern |

### Methods

| n/a | Involved in the study |
|---|---|
| ☒ | ☐ ChIP-seq |
| ☒ | ☐ Flow cytometry |
| ☒ | ☐ MRI-based neuroimaging |

## Antibodies

| | |
|---|---|
| Antibodies used | SMAC (MBL, JM-3298-100) dilution 1:1000, STARD7 (Proteintech 15689-1-AP) dilution 1:2500, FLAG (WAKO 018-22381) dilution 1:1000, SDHA (Abcam Ab14715) dilution 1:10000, TRANSFERRIN (Invitrogen 13-6800) dilution 1:1000, CLPP (Sigma HPA010649) dilution 1:1000, YME1L (Proteintech 11510-1-AP) dilution 1:1000, MIC60 (Nobus Biologicals 100-1919) dilution 1:1000, VDAC2 (Proteintech 11663-1-AP) dilution 1:1000. PARL antibody is defined previously [1], dilution 1:1000 |
| Validation | Antibodies STARD7 (Extended data fig. 1e) and PARL (Extendend data fig. 1c) are validated by observing no bands in the knockout cell lines of these proteins. FLAG antibody was validated by using overexpressed Flag-tagged proteins (Extended data fig. 1c). SMAC and SDHA antibody has been validated before in our lab (Saita et al., 2017 Nature Cell Biology). YME1l has been validated before in our lab (MacVicar et al., 2019 Nature). Transferrin, CLPP, antibody has been verified by the manufacturer using knockdown experiments. MIC60 validated in PMID: 34037656 according to manufacturer. VDAC2 vaildated by manufacturer using immunoprecipitation here https://www.ptglab.com/products/VDAC2-Antibody-11663-1-AP.htm. |

# Eukaryotic cell lines

Policy information about cell lines

| | |
|---|---|
| Cell line source(s) | HCT116: ATCC, catalog number: CCL-247. HeLa (CCL-2) cells were purchased from ATCC. HEK293T cells were described in PMID: 31695197 but only used for virus production and no biological interpretations were based on HEK293T cells.  Further information regarding generation of knockout cell line details are included in the Methods section. |
| Authentication | Cell lines were not authenticated |
| Mycoplasma contamination | All the cell lines were routinely tested for mycoplasma and only when tested negative were used for the experiments. |
| Commonly misidentified lines (See ICLAC register) | No commonly misidentified lines were used in this study |

# Animals and other organisms

Policy information about studies involving animals; ARRIVE guidelines recommended for reporting animal research

| | |
|---|---|
| Laboratory animals | C57BL/6N mice, male 5-weeks old mice were used. Mice were maintained at the specific-pathogen-free animal facility of the Cologne Excellence Cluster on Cellular Stress Responses in Aging-Associated Diseases (CECAD) Research Centre with 12 h light cycle and regular chow diet. |
| Wild animals | No Wildtype animals were used in the study. |
| Field-collected samples | No field collected samples were used in this study. |
| Ethics oversight | All animal experiments were approved by Landesamt für Natur, Umwelt und Verbraucherschutz Nordrhein-Westfalen, Germany and the Cologne Excellence Cluster on Cellular Stress Responses in Aging-Associated Diseases (CECAD) mouse facility regulations. |

Note that full information on the approval of the study protocol must also be provided in the manuscript.

