## [Peer Review File · Nature Cell Biology]

Peer Review Information

Journal: Nature Cell Biology

Manuscript Title: Mitochondria regulate intracellular coenzyme Q transport and ferroptotic resistance via STARD7

Corresponding author name(s): Professor Thomas Langer

Editorial Notes:

Reviewer Comments & Decisions:

Decision Letter, initial version:

Dear Thomas,

Thank you for submitting your manuscript, "Mitochondria regulate intracellular coenzyme Q transport and ferroptotic resistance via STARD7", to Nature Cell Biology. It has now been seen by 3 referees,

who are experts in ferroptosis (Referee #1); mitochondrial metabolism (Referee #2); and CoQ/electron transport chain (Referee #3). As you will see from their comments (attached below), they found this work of potential interest but have raised substantial concerns, which in our view would need to be addressed with considerable revisions before we can consider publication in Nature Cell Biology.

As per our standard editorial process, we have now discussed the referee reports in detail within the editorial team, including with our Chief Editor, to identify key referee points that should be addressed with priority to strengthen the core conclusions of the study. To guide the scope of the revisions, I have listed these points below. Our typical revision period is 6 months; we are committed to providing a fair and constructive peer-review process, so please feel free to contact me if you would like to discuss any of the referee comments further or if you anticipate any delays or issues addressing the reviews.

Despite interest and enthusiasm, in particular from Revs#2-3, we found the reviewers' remarks significant. In our view, the referees' concerns regarding the strength of the model and the evidence that STARD7 directly affects CoQ transport would need to be addressed with experiments and data, and possible mechanisms of action should be explored as guided by the reviewers' points. Such insight would strengthen the study and proposed model. Reconsideration of the study for this journal and re-engagement of referees will depend on the strength of these revisions. The reviewers provided suggestions that we feel are constructive to bolster the conclusions and we encourage you to dedicate rigorous revision efforts to address the reviews, in particular:

A- please strengthen the evidence that cytosolic STARD7 acts to transport CoQ and test the reviewers' suggestions to explore the underlying mechanism: Rev#1 points #1-2, Rev#3 point #6. Please also rule out alternative models, i.e., Rev#1 points #3-4.

B- please address all requests for controls and comments about the different impacts of cyto-STARD7 vs mito-STARD7 constructs: Rev#2 points #2, #4; Rev#3 #1-2-3-5

C- please address the referees' questions about the relationship between PC transport and CoQ transport by STARD7: Rev#2 points #3-4, Rev#3 #3.

D- All other referee concerns pertaining to strengthening existing data, methodological details, clarifications and textual changes, and in particular the minor points, should also be addressed.

E- Finally, please pay close attention to our guidelines on statistical and methodological reporting (listed below) as failure to do so may delay the reconsideration of the revised manuscript. In particular please provide:

- a Supplementary Table including all numerical source data in Excel format, with data for different figures provided as different sheets within a single Excel file. The file should include source data giving rise to graphical representations and statistical descriptions in the paper and for all instances where the figures present representative experiments of multiple independent repeats, the source data of all

repeats should be provided.

We would be happy to consider a revised manuscript that would satisfactorily address these points, unless a similar paper is published elsewhere or is accepted for publication in Nature Cell Biology in the meantime.

- ensure that it conforms to our format instructions and publication policies (see below and <https://www.nature.com/nature/for-authors>).
- provide a point-by-point rebuttal to the full referee reports verbatim, as provided at the end of this letter.
- provide the completed Reporting Summary (found here <https://www.nature.com/documents/nr-reporting-summary.pdf>). This is essential for reconsideration of the manuscript will be available to editors and referees in the event of peer review. For more information see <http://www.nature.com/authors/policies/availability.html> or contact me.

When submitting the revised version of your manuscript, please pay close attention to our [href="https://www.nature.com/nature-portfolio/editorial-policies/image-integrity">Digital Image Integrity Guidelines](https://www.nature.com/nature-portfolio/editorial-policies/image-integrity). and to the following points below:

Nature Cell Biology is committed to improving transparency in authorship. As part of our efforts in this direction, we are now requesting that all authors identified as 'corresponding author' on published papers create and link their Open Researcher and Contributor Identifier (ORCID) with their account on the Manuscript Tracking System (MTS), prior to acceptance. ORCID helps the scientific community achieve unambiguous attribution of all scholarly contributions. You can create and link your ORCID from the home page of the MTS by clicking on 'Modify my Springer Nature account'. For more information please visit www.springernature.com/orcid.

This journal strongly supports public availability of data. Please place the data used in your paper into a public data repository, or alternatively, present the data as Supplementary Information. If data can only be shared on request, please explain why in your Data Availability Statement, and also in the correspondence with your editor. Please note that for some data types, deposition in a public

repository is mandatory - more information on our data deposition policies and available repositories appears below.

[REDACTED]

We hope that you will find our referees' comments and editorial guidance helpful. Please do not hesitate to contact me if there is anything you would like to discuss. Thank you again very much for considering NCB for your work.

Best wishes,

Melina

Melina Casadio, PhD
Senior Editor, Nature Cell Biology
ORCID ID: <https://orcid.org/0000-0003-2389-2243>

Reviewers' Comments:

Reviewer #1:

Remarks to the Author:

PARL is a member of the rhomboid family of intramembrane cleaving peptidases, which is required for the maturation of several important mitochondrial proteins and thus has an essential physiological role in the maintenance of mitochondrial structure and function. Previous study showed that loss of PARL caused CoQ depletion in multiple mouse tissues. In the current study, Deshwal et al. confirmed this finding and further showed that one of the PARL substrates, STARD7, restored CoQ in PARL-deficient cells.

STARD7 is a member of the START (StAR-related lipid transfer) domain-containing protein family. STARD7 localizes both to the cytosol and the mitochondrial intermembrane space (IMS), where it serves as an intramitochondrial lipid transfer protein of phosphatidic choline (PC). The loss of STARD7 impairs the accumulation of PC in mitochondrial membranes, mitochondrial ATP production, and cristae morphogenesis. The authors previously reported that PARL-mediated cleavage of STARD7 during STARD7 mitochondrial import is required for STARD7's release from mitochondria to the cytosol.

In the current study, the authors present data to support their conclusion that mito STRAD7 is required for CoQ synthesis, while cyto STRAD7 is required for the transport of CoQ from mitochondria

to cytosol, and therefore is important for the cellular protection to ferroptosis, a cell death process caused by iron-dependent phospholipid peroxidation. This hypothesis is interesting in that it proposes new regulators for the previously established function of CoQ synthesis in ferroptosis. However, the current experimental data failed to provide sufficient support for this hypothesis. Many questions need to be addressed.

1. The evidence of cyto-STARD7 as a cyto CoQ transporter is rather weak. First of all, the comparison of Mito-CoQ (fraction 1) and PM-CoQ (fraction 3) is not convincing. As seen in Fig 4f, fraction 3 only represent a minority of PM. The fractionation protocol needs to be optimized. Second, some biochemistry assays are absolutely required for measuring the affinity of STARD7 to CoQ, as well as the transfer activity of STARD7 (see the Ref: Horibata, et al. JBC 2010).

2. The cyto-STARD7 construct lacks the MTS, which means the ectopically expressed cyto-STARD7 will not be imported to mitochondria after its synthesis. As such, it is hard to imagine how cyto-STARD7 can transport CoQ from the mitochondria to the cytosol. This cyto-STARD7 construct did not mimic the STARD7 release process. Actually, the overexpression of cyto-STARD7 did not increase PM-CoQ in STARD7 $-/-$ cells (Fig 4g), although cyto-STARD7 did slightly increase PM-CoQ in WT cells (log₂FC around 0.3 and 0.6). In any case, to demonstrate that the release of STARD7 from mitochondria is responsible for CoQ transport to the cytosol, additional conditions need to be tested, e.g. compare PM-CoQ vs Mito-CoQ in wt vs PARL $-/-$ cell, in ctrl siRNA vs TIMM23 siRNA cells (TIMM23 RNAi should promote the release of STARD7), etc.

3. Loss of PARL decreased CoQ synthesis. A simple and alternative explanation, without going into the CoQ translocation hypothesis of the authors (as it is not well supported by their data), is that the expression of multiple proteins in CoQ synthesis pathway might be downregulated. Is this the case and can overexpression of mito-STARD7 rescue CoQ synthesis in PARL $-/-$ cells through restoring the expression of these proteins?

4. The connection of CoQ to ferroptosis is clear, and thus the only novelty of this study is the mechanistic connection of PARL-STARD7 to CoQ translocation, which is not convincing nevertheless. Since loss of PARL or STARD7 will significantly affect mitochondria function, one cannot rule out that PARL-STARD7 alteration indirectly impacts mitochondrial CoQ synthesis via disrupting mitochondrial function. Further, as PARL-STARD7 regulates CoQ synthesis in tissue specific manner (Fig B vs Fig C, also see Ref 30), they need to assess the significance of PARL-STARD7-CoQ-Ferroptosis pathway in 2~3 cell lines of different tissue origin (cell death as well as CoQ measurement). Last but not the least, as mentioned before, cyto-STARD7 construct is not a suitable control as it is completely unrelated to mitochondrial translocation.

Minor points:

1. For the cell death analysis, why sometimes use cell death area/phase area, sometimes dead cell count/phase area?

2. Why use Erastin + PUFA to induce ferroptosis? This condition is not common. More common inducing conditions are needed.

3. To confirm CoQ is required for STARD7's ferroptosis suppression function, direct CoQ synthesis inhibition (e.g. CoQ2 KO, 4-CBA) should be tested, in addition to iFSP1.

Reviewer #2:
Remarks to the Author:
Overview

This is a strong paper that ties together protein processing, CoQ transport and ferroptosis to generate a nice story that is supported by the data. I have mainly minor comments.

Major points

1 The focus of the paper is on the link between CoQ distribution within the cell and ferroptosis. While STARD7 is clearly involved in CoQ transport, the mechanism remains obscure. The authors discuss the difficulty of extracting such a hydrophobic molecule from the inner membrane, getting it from the mito IM and then to the mito OM and then -rightly in my view - suggest that extracting it to a transport protein and reinsert it into the plasma membrane is likely to require an extensive protein machinery. The authors could expand on this - there have been a few attempts to identify the putative machinery that moves CoQ from mitos to the plasma membrane (eg FEBS Journal (2010) 277 2067-2082. Free Radical Biology and Medicine (2020) 154 105-118).

2 The measurements of CoQ9/10 distribution between the mitos and plasma membrane is not easy, but the data are presented as fold changes, which may have been done because of the high variability I'd expect in such an analysis. I really would like to see the actual amounts measured in the compartments normalised to protein or some other marker. It would also be good to measure the whole cell CoQ9/10 content and to see what proportion of the total pool is in the mito and plasma membranes.

3 The mechanism of the selectivity for loss of CoQ biosynthesis machinery in the IM but not of OX phos was intriguing. The implication is that this is secondary to PC transport - was it confirmed that expression of mito-STARD7 restored mitochondrial IM PC levels? Also, did loss of STARD7 affect cristae morphology and was this reversed by mito-STARD7? Can you replicate this by altering PC synthesis?

4 The authors inferred in Fig 5 that overexpression of cyto-STARD7 affected respiration by extracting CoQ from the IM. I couldn't see that loss of CoQ was actually measured and whether other interpretations are possible? Did cyto-STARD7 overexpression affect the CoQ biosynthetic machinery or OX PHOS complexes? Could overexpression of cyto-STARD7 have disrupted/sequestered PC and thereby indirectly affected mitochondria? This section was the only weak-ish section in the paper and the authors might want to consider strengthening or removing it.

Minor points

1 Fig 1g shows cleavage of STAD7 by PARL with the whole cleaved protein temporarily in the IMS. Fig5e shows this more accurately with cleavage at the IM by PARL while the cytosolic module is still in the cytosol, precluding the need for a mysterious mechanism for the cleaved STAD7 to leave the IMS, implied in Fig 1g. I suggest replace Fig 1g with something more like Fig 5e.

Reviewer #3:

Remarks to the Author:

In this article the authors discover a connection between the proteolytic activity of PARL, a mitochondrial rhomboid protease and the functional role of one of its polypeptide substrates, STARD7 in controlling transport of coenzyme Q (CoQ or ubiquinone). The authors show that PARL regulates both the synthesis and the cellular distribution of CoQ via its ability to proteolytically process STARD7, which changes its "zip code", and relocates STARD7 from the mitochondria to the cytosol. Although PARL was previously known to affect the content of CoQ, the mechanism(s) responsible remained mysterious. This work unmasks the actions of PARL that affect both CoQ synthesis within the mitochondria, as well as the transport of this aqueous insoluble lipid to non-mitochondrial destinations within the cell. Transport of CoQ from mitochondria to the plasma membrane plays an important role in protecting cells from cell death due to lipid peroxidation, termed ferroptosis.

I believe that the paper could be enhanced if the authors addressed the following points:

1. The authors construct variants of STARD7 that accumulate exclusively in the mitochondrial IMS (mito-STARD7) or in the cytosol (cyto-STARD7, that lacks the mitochondrial targeting sequence). However, it is important for the authors to determine experimentally that the said STARD7 variants are in fact exclusively located within these distinct compartments.
2. Additionally, it might be interesting to express STARD7 that is targeted to the Mito inner membrane, but that is not able to be cleaved by PARL, as this would presumably prevent any retrograde relocation of processed STARD7 from the IMS to the cytosol. This experiment would also address the author's claim that unprocessed STARD7 is present in the inner membrane of PARL^{-/-} cells but is not sufficient to ensure the accumulation of CoQ, as stated on page 6.
3. In Figure 1j (and Supplementary Figure 1g) cells expressing the mutant STARD7R189Q appear to have a decreased content of CoQ10 (and CoQ9) relative to the STARD7 null mutant. This result seems surprising – that is, why would the point mutant have even lower CoQ content than the null mutant? The STARD7R189Q mutant is known to affect the binding of phosphatidyl choline (PC). Presumably the null mutant would also have a defect in binding PC. The authors conclude that maintenance of CoQ levels depend on the PC transport. Yet the result suggests that expression of STARD7R189Q has a more negative effect on CoQ content than does the null. Could it be that the STARD7R189Q mutant polypeptide has acquired an additional activity or a type of dominant negative effect that further decreases CoQ content relative to the null mutant?
5. The authors convincingly demonstrate that suppression of ferroptosis (death by lipid peroxidation) depends on both mitochondrial and cytosolic STARD7, and they conclude that suppression of ferroptosis depends on STARD7 being present in both the IMS and the cytosol. The text in this section emphasizes that the addition of PUFA (arachidonic acid) would be expected to induce the lipid peroxidation in the plasma membrane (and so lead to cell death). However, it seems likely that addition of PUFA will impose lipid peroxidation on all intracellular membranes, including the mitochondrial membranes. In fact, recent literature (doi: 10.1038/s41586-021-03539-7.) suggests that the action of mitochondrial dihydroorotate dehydrogenase protects cells against PUFA-mediated ferroptosis via its ability to convert mitochondrial CoQ to CoQH₂. The results in the present manuscript support this interpretation, and suggest that CoQH₂ plays an important role as a chain breaking antioxidant in mitochondrial as well as the plasma membrane. Perhaps this could at least partially explain why the suppression of ferroptosis depends on both the mitochondrial and cytosolic forms of STARD7.

6. Fig 5 depicts a model for the action of STARD7. Yet, the “squiggle” representing STARD7 is not terribly illuminating as to its mode of action. Is it possible that the START domain present in STARD7 might be able to bind CoQ as a ligand? The authors state on page 13 that, “...a complete membrane extraction of CoQ and binding with the lipid binding cavity of STAFD7 appears unlikely give steric constraints and the hydrophobicity of the polyprenoid tail of CoQ.” What is the basis for this statement? Has modeling of CoQ within the hydrophobic START domain of STARD7 been attempted? STARD7 has a hydrophobic pocket that binds PC... is it possible that this pocket could “trade PC” for CoQ? This is how alpha-tocopherol transfer protein is postulated to interact with phosphatidylinositols and traffic vitamin E, so there is at least precedent for the idea that a phospholipid may “direct” the trafficking of lipids containing polyisoprenoid-based moieties.

General comments

1. It can be confusing to refer to “reduced” levels of CoQ. This is because reduced can mean either decreased, or alternatively, can indicate that the CoQ is reduced (e.g. CoQ is reduced to the CoQH2 hydroquinone). It would be helpful for the authors to peruse their text for this usage, and to replace the word “reduced” with “decreased” whenever the intent is to indicate a decreased content of CoQ.
2. There are two isoforms of human STARD7, STARD7-I and STARD7-II. How do these two isoforms relate to the PARL mediated processing reported here? This could be important since the processing by PARL may differ depending on the isoform being expressed.

AUTHOR AFFILIATIONS – should be denoted with numerical superscripts (not symbols) preceding the

names. Full addresses should be included, with US states in full and providing zip/post codes. The corresponding author is denoted by: "Correspondence should be addressed to [initials]."

Methods should be written concisely, but should contain all elements necessary to allow interpretation and replication of the results. As a guideline, Methods sections typically do not exceed 3,000 words. The Methods should be divided into subsections listing reagents and techniques. When citing previous methods, accurate references should be provided and any alterations should be noted. Information must be provided about: antibody dilutions, company names, catalogue numbers and clone numbers for monoclonal antibodies; sequences of RNAi and cDNA probes/primers or company names and catalogue numbers if reagents are commercial; cell line names, sources and information on cell line identity and authentication. Animal studies and experiments involving human subjects must be reported in detail, identifying the committees approving the protocols. For studies involving human

subjects/samples, a statement must be included confirming that informed consent was obtained. Statistical analyses and information on the reproducibility of experimental results should be provided in a section titled "Statistics and Reproducibility".

All Nature Cell Biology manuscripts submitted on or after March 21 2016 must include a Data availability statement as a separate section after Methods but before references, under the heading "Data Availability". For Springer Nature policies on data availability see <http://www.nature.com/authors/policies/availability.html>; for more information on this particular policy see <http://www.nature.com/authors/policies/data/data-availability-statements-data-citations.pdf>. The Data availability statement should include:

- Accession codes for primary datasets (generated during the study under consideration and designated as "primary accessions") and secondary datasets (published datasets reanalysed during the study under consideration, designated as "referenced accessions"). For primary accessions data should be made public to coincide with publication of the manuscript. A list of data types for which submission to community-endorsed public repositories is mandated (including sequence, structure, microarray, deep sequencing data) can be found here <http://www.nature.com/authors/policies/availability.html#data>.
- Unique identifiers (accession codes, DOIs or other unique persistent identifier) and hyperlinks for datasets deposited in an approved repository, but for which data deposition is not mandated (see here for details <http://www.nature.com/sdata/data-policies/repositories>).
- At a minimum, please include a statement confirming that all relevant data are available from the authors, and/or are included with the manuscript (e.g. as source data or supplementary information), listing which data are included (e.g. by figure panels and data types) and mentioning any restrictions on availability.
- If a dataset has a Digital Object Identifier (DOI) as its unique identifier, we strongly encourage including this in the Reference list and citing the dataset in the Methods.

We recommend that you upload the step-by-step protocols used in this manuscript to the Protocol Exchange. More details can found at www.nature.com/protocolexchange/about.

All imaging data should be accompanied by scale bars, which should be defined in the legend. Cropped images of gels/blots are acceptable, but need to be accompanied by size markers, and to retain visible background signal within the linear range (i.e. should not be saturated). The boundaries of panels with low background have to be demarked with black lines. Splicing of panels should only be

considered if unavoidable, and must be clearly marked on the figure, and noted in the legend with a statement on whether the samples were obtained and processed simultaneously. Quantitative comparisons between samples on different gels/blots are discouraged; if this is unavoidable, it should only be performed for samples derived from the same experiment with gels/blots were processed in parallel, which needs to be stated in the legend.

The total number of Supplementary Figures (not including the “unprocessed scans” Supplementary Figure) should not exceed the number of main display items (figures and/or tables (see our Guide to Authors and March 2012 editorial <http://www.nature.com/ncb/authors/submit/index.html#suppinfo>; <http://www.nature.com/ncb/journal/v14/n3/index.html#ed>). No restrictions apply to Supplementary Tables or Videos, but we advise authors to be selective in including supplemental data.

GUIDELINES FOR EXPERIMENTAL AND STATISTICAL REPORTING

REPORTING REQUIREMENTS – We are trying to improve the quality of methods and statistics reporting in our papers. To that end, we are now asking authors to complete a reporting summary

that collects information on experimental design and reagents. The Reporting Summary can be found here <https://www.nature.com/documents/nr-reporting-summary.pdf>) If you would like to reference the guidance text as you complete the template, please access these flattened versions at <http://www.nature.com/authors/policies/availability.html>.

Author Rebuttal to Initial comments
--

Reply to the summary points of the editor:

A- please strengthen the evidence that cytosolic STARD7 acts to transport CoQ and test the reviewers' suggestions to explore the underlying mechanism: Rev#1 points #1-2, Rev#3 point #6. Please also rule out alternative models, i.e., Rev#1 points #3-4.

To strengthen the evidence that cytosolic STARD7 is directly involved in CoQ transport, we used reconstitution experiments to demonstrate that purified STARD7 can bind to CoQ4 and extract it from liposomes *in vitro* (Rev#1 points #1-2; Rev#2 point #1). A mutation in the lipid binding cavity of STARD7 is shown to impair CoQ4 binding. These experiments also revealed an intriguing competition between PC and CoQ4 for STARD7 binding, in line with a speculation of Rev#3 (point 6). These experiments are shown in the new Fig. 5. Moreover, we exclude alternative models by showing

- a) that PARL deletion in HeLa cells does not affect the CoQ biosynthetic machinery *per se* (Rev#1 point 3) (Reviewer Fig. 1).
- b) that cells with normal mitochondrial function and CoQ levels but lacking cytosolic STARD7 (*STARD7*^{-/-} cell expressing mito-STARD7) are sensitive to ferroptosis, excluding indirect effects of mitochondrial deficiencies (Rev#1 point 4).

B- please address all requests for controls and comments about the different impacts of cyto-STARD7 vs mito-STARD7 constructs: Rev#2 points #2, #4; Rev#3 #1-2-3-5

To comply with Rev#2 points #2 and #4, we added the raw data showing consistent effects for all replicates (Extended Data Fig. 4b). We also show CoQ levels in plasma membrane and mitochondrial fractions (Fig. 4) and exclude indirect effects of cyto-STARD7 expression on the steady state levels of CoQ biosynthetic enzymes and OXPHOS complexes (Extended Data Fig. 6a).

We have extensively studied the localization of mito-STARD7 and cyto-STARD7 in our previous work (Saita et al., 2018). We now also provide additional evidence for the localization of cyto-STARD7 in WT cells (Rev#3 point 1 and 2) (Extended Data Fig. 3b). Moreover, we agree with the reviewer's comment (Rev#3 point 5) that mito-STARD7 requirement to protect cells against ferroptosis might be partially explained via DHODH-mediated CoQ reduction and discuss this in the text.

C- please address the referees' questions about the relationship between PC transport and CoQ transport by STARD7: Rev#2 points #3-4, Rev#3 #3.

Our reconstitution experiments revealed competitive binding of PC and CoQ4 to the lipid binding cavity of STARD7 (Fig. 5d) (Rev#3). We show that cyto-STARD7 overexpression does not affect cellular growth or total CoQ levels, but decreases mitochondrial CoQ content by increasing the export of CoQ from the mitochondria to

the plasma membrane. This decreased mitochondrial CoQ pool impairs respiratory growth on galactose-containing medium (Rev#2 point #4). We have previously shown that expression of mito-STARD7 in *STARD7*^{-/-} cells maintains mitochondrial PC levels, OXPHOS activity and cristae morphogenesis (Saita et al., 2018) (Rev#2 point #3).

D- All other referee concerns pertaining to strengthening existing data, methodological details, clarifications and textual changes, and in particular the minor points, should also be addressed.

We have strengthened existing data by 1) complementary reconstitution experiments demonstrating CoQ binding to STARD7 *in vitro* (Fig. 5), 2) by showing that CoQ is required for the protective effect of cyto-STARD7 (which is lost upon inhibition of CoQ synthesis) (Fig. 3h, i, j); 3) by excluding indirect effects caused by altered OXPHOS or CoQ biosynthetic enzymes (Extended Data Fig. 6a), and 4) by demonstrating similar roles of PARL and STARD7 on CoQ levels and ferroptotic resistance in another cell line (HCT116)(Fig. 2g, h, f). We have improved the manuscript text and addressed all minor points.

E- Finally, please pay close attention to our guidelines on statistical and methodological reporting (listed below) as failure to do so may delay the reconsideration of the revised manuscript. In particular please provide:

We have represented all the data in the manuscript as 95% confidence interval for the mean to show the statistical significance between the groups.

We have provided the unprocessed images of all the blots in a multi-page pdf file as a supplementary figure. The sections presented in the figures are highlighted in red boxes.

We have provided all source data in an excel sheet with all independent replicates as a supplementary table.

Reply to reviewer #1:

PARL is a member of the rhomboid family of intramembrane cleaving peptidases, which is required for the maturation of several important mitochondrial proteins and thus has an essential physiological role in the maintenance of mitochondrial structure and function. Previous study showed that loss of PARL caused CoQ depletion in multiple mouse tissues. In the current study, Deshwal et al. confirmed this finding and further showed that one of the PARL substrates, STARD7, restored CoQ in PARL-deficient cells.

STARD7 is a member of the START (StAR-related lipid transfer) domain-containing protein family. STARD7 localizes both to the cytosol and the mitochondrial intermembrane space (IMS), where it serves as an intramitochondrial lipid transfer protein of phosphatidic choline (PC). The loss of STARD7 impairs the accumulation of PC in mitochondrial membranes, mitochondrial ATP production, and cristae morphogenesis. The authors previously reported that PARL-mediated cleavage of STARD7 during STARD7 mitochondrial import is required for STARD7's release from mitochondria to the cytosol.

In the current study, the authors present data to support their conclusion that mito STARD7 is required for CoQ synthesis, while cyto STARD7 is required for the transport of CoQ from mitochondria to cytosol, and therefore is important for the cellular protection to ferroptosis, a cell death process caused by iron-dependent phospholipid peroxidation. This hypothesis is interesting in that it proposes new regulators for the previously established function of CoQ synthesis in ferroptosis. However, the current experimental data failed to provide sufficient support for this hypothesis. Many questions need to be addressed.

We thank the reviewer for their careful assessment of our previous work. We would like to emphasize, however, that we consider as the most novel and unexpected finding in our manuscript the identification of STARD7 as the very first component required for the cellular distribution of CoQ, providing first insight into a completely unknown cellular transport pathway. Moreover, we demonstrate that cell proliferation and survival depend on the coordinated distribution of CoQ between mitochondria and the plasma membrane, which is ensured by the dual localization of STARD7 to both the IMS and the cytosol and which is regulated by PARL.

As outlined in more detail below, our cumulative data now provide compelling evidence for the requirement of STARD7 for the distribution of CoQ translocation to the plasma membrane:

- Loss of STARD7 exclusively in the cytosol (in the presence of mitochondrial STARD7, normal mitochondrial function and normal CoQ synthesis) leads to decreased CoQ levels in the plasma membrane and increased vulnerability to ferroptosis.
- Overexpression of cyto-STARD7 (in the presence of mitochondrial STARD7 and normal CoQ synthesis) is sufficient to increase CoQ levels in the plasma membrane and ferroptotic resistance of the cells.
- Purified, mature STARD7 binds directly to CoQ4 *in vitro*.

1. *The evidence of cyto-STARD7 as a cyto CoQ transporter is rather weak. First of all, the comparison of Mito-CoQ (fraction 1) and PM-CoQ (fraction 3) is not convincing. As seen*

in Fig 4f, fraction 3 only represent a minority of PM. The fractionation protocol needs to be optimized. Second, some biochemistry assays are absolutely required for measuring the affinity of STARD7 to CoQ, as well as the transfer activity of STARD7 (see the Ref: Horibata, et al. JBC 2010).

The reviewer raises two concerns that we have carefully addressed:

- a) To comply with the request of the reviewer, we have extended our cell fractionation experiments in cells expressing cyto-STARD7 and in wildtype cells. We now also show the CoQ levels in an additional cellular fraction containing proteins of the plasma membrane (fraction 2 in Fig. 4; Reviewer Fig. 1). Similar to fraction 3 (PM fraction in Fig. 4), we observed increased CoQ levels in this plasma membrane-enriched fraction of cyto-STARD7 expressing cells, corroborating our conclusion that cyto-STARD7 supports CoQ transport to the plasma membrane (Reviewer Fig. 1). Although the analysis of fraction 2 and 3 shows similar effects on CoQ levels, we prefer not to include these data in the manuscript, as the contamination with mitochondrial proteins is higher in fraction 2 than in fraction 3 as demonstrated by our proteomic analysis (see Fig. 4b and c). We would like to stress that we are highly confident of these cell fractionation experiments, since we did not rely only on individual marker proteins (detected by immunoblotting) but assessed the purity of each fraction by MS-based proteomics. In our opinion, this sets a new standard for cell fractionation experiments.

Reviewer Fig 1: CoQ levels in different cellular fractions. Difference in the abundance of CoQ10, CoQ9 and PC levels in fraction 1, 2 and 3 in cyto-STARD7 overexpressing cells relative to WT.

Since we aimed to compare CoQ distribution and relative CoQ levels between mitochondria and the plasma membrane, we extracted and quantified CoQ from the identical cellular lysate. It is expected that the final fraction only contains part of plasma membrane proteins and lipids (as also illustrated by the distribution of plasma membrane proteins between fraction 2 and 3).

Finally, we would like to emphasize that we have performed cellular fractionation to compare the distribution of CoQ between wildtype and cyto-STAR7 expressing cells, not to determine absolute CoQ levels in different cellular compartments. Although interesting, the latter question requires to consider also other cellular membranes, which is outside the focus of the present studies. In response to Rev#2, we now also include data showing that ~90% of CoQ resides in mitochondria, while ~10% are found in non-mitochondrial membranes (Fig. 4d, e).

- a) We agree with the reviewer that it is of utmost importance to understand how STAR7 affects CoQ distribution, although this might be complex and involve multiple other proteins. To address this issue, we have examined a possible direct interaction of STAR7 with CoQ. Purified STAR7 was incubated with liposomes which contain CoQ4, CoQ9 or CoQ10 and which vary in their phospholipid compositions. We then isolated STAR7 and analyzed associated lipids that were extracted from liposomes by STAR7 by mass spectrometry (Fig. 5). This approach revealed direct binding of CoQ4 to STAR7, demonstrating that STAR7 directly affects CoQ distribution. We did not observe CoQ9 and CoQ10 binding to STAR7. This likely relates to their hydrophobicity, which may prevent their extraction from liposomes or their recovery upon STAR7 purification as discussed in the manuscript. Interestingly, CoQ binding to STAR7 was only observed in the absence of DOPC in the liposomes, indicating competition between both STAR7-binding lipids (as also speculated by Rev#3). Increasing the concentration of DOPC decreased CoQ4 binding to STAR7, highlighting the specificity of this interaction (Fig. 5c). The specificity of CoQ4 binding was further corroborated by mutating R189, which is located in the binding cavity of STAR7 and required for PC binding to STAR7 (Extended Fig. 5c; Bockelmann et al., 2018). The mutation R189Q abolished PC binding and strongly reduced CoQ4 binding (Extended Fig. 5c; Fig. 5d), providing additional support for the specificity of CoQ4 binding to STAR7. Although the likely complex machinery involved in CoQ distribution deserves further experimentation in the future, our results demonstrate that STAR7 binds CoQ4 and thus directly affects CoQ transport.

2. The cyto-STAR7 construct lacks the MTS, which means the ectopically expressed cyto-STAR7 will not be imported to mitochondria after its synthesis. As such, it is hard to imagine how cyto-STAR7 can transport CoQ from the mitochondria to the cytosol. This cyto-STAR7 construct did not mimic the STAR7 release process. Actually, the overexpression of cyto-STAR7 did not increase PM-CoQ in STAR7 -/- cells (Fig 4g), although cyto-STAR7 did slightly increase PM-CoQ in WT cells (log2FC around 0.3 and 0.6). In any case, to demonstrate that the release of STAR7 from mitochondria is responsible for CoQ transport to the cytosol, additional conditions need to be tested, e.g. compare PM-CoQ vs Mito-CoQ in wt vs PARL-/- cell, in ctrl siRNA vs TIMM23 siRNA cells (TIMM23 RNAi should promote the release of STAR7), etc.

There is apparently some misunderstanding and we apologize if we have not clearly described our model in the original manuscript. We do not propose that STARD7 import into and release from mitochondria (upon PARL cleavage) are coupled to CoQ transport to the plasma membrane. We have no experimental evidence supporting such a model and, as pointed out by the reviewer, consider it also as unlikely, as STARD7 is released upon PARL cleavage during import likely in an unfolded conformation, which does not allow specific lipid binding/loading (Saita et al., 2018). Rather, we demonstrate, first, that the dual localization of STARD7 both in the IMS and the plasma membrane is a prerequisite for CoQ distribution and ferroptotic resistance, and, second, that the unbalanced accumulation of cyto-STARD7 relative to mito-STARD7 (upon cyto-STARD7 overexpression) impairs OXPHOS function.

Although we have previously shown that downregulation of TIMM23 affects the distribution of STARD7 between the IMS and the cytosol (Saita et al., 2018), this approach is not suitable to examine the effect on cellular CoQ distribution and ferroptotic resistance, since depletion of TIMM23, an essential constituent of the TIM23 complex, broadly affects mitochondrial functions (including import of CoQ biosynthetic enzymes).

As pointed out by the reviewer, we did not observe increased CoQ levels in the plasma membrane upon overexpression of cyto-STARD7 in *STARD7*^{-/-} cells (Fig. 4g). Consistently, cyto-STARD7 expression did not protect *STARD7*^{-/-} cells against ferroptosis (Fig. 2i, j). This is expected and explained by the lack of mito-STARD7 in these cells, which is required to preserve CoQ synthesis in *STARD7*^{-/-} cells (Fig. 1i). Only in the presence of both mito- and cyto-STARD7 CoQ is normally synthesized and transported to the plasma membrane (see above).

Finally, we would like to stress that cyto-STARD7 is identical at the amino acid level to mature STARD7 released from mitochondria and therefore allows to assess the function of cytosolic STARD7. Exogenous expression of STARD7 specifically in the cytosol of wildtype cells allowed us to demonstrate that cyto-STARD7 is a limiting factor for CoQ distribution and ferroptotic resistance.

3. Loss of PARL decreased CoQ synthesis. A simple and alternative explanation, without going into the CoQ translocation hypothesis of the authors (as it is not well supported by their data), is that the expression of multiple proteins in CoQ synthesis pathway might be downregulated. Is this the case and can overexpression of mito-STARD7 rescue CoQ synthesis in PARL^{-/-} cells through restoring the expression of these proteins?

To address this comment of the reviewer, we determined the cellular proteome of wildtype and *PARL*^{-/-} cells. This analysis revealed alterations in the proteome in the absence of PARL (as expected) but demonstrated that the steady state level of proteins of the CoQ synthesis pathway are not decreased (Reviewer Fig. 2). Since we have published the mitochondrial proteome of *PARL*^{-/-} cells previously (Saita et al., 2017), we prefer not to include this figure in the manuscript. However, we show

now the unaltered accumulation of CoQ biosynthetic enzymes (and OXPHOS subunits) in cells overexpressing cyto-STAR7 in the Extended Data Fig. 6a.

Reviewer Fig 2: Steady state levels of CoQ biosynthetic enzymes in *PARL*^{-/-} HeLa cells. Volcano plot showing the whole proteome of WT and *PARL*^{-/-} HeLa cells. Significantly changed proteins are highlighted in blue and grey represents unchanged proteins. CoQ biosynthetic proteins are represented in red.

4. *The connection of CoQ to ferroptosis is clear, and thus the only novelty of this study is the mechanistic connection of PARL-STAR7 to CoQ translocation, which is not convincing nevertheless. Since loss of PARL or STAR7 will significantly affect mitochondria function, one cannot rule out that PARL-STAR7 alteration indirectly impacts mitochondrial CoQ synthesis via disrupting mitochondrial function. Further, as PARL-STAR7 regulates CoQ synthesis in tissue specific manner (Fig B vs Fig C, also see Ref 30), they need to assess the significance of PARL-STAR7-CoQ-Ferroptosis pathway in 2~3 cell lines of different tissue origin (cell death as well as CoQ measurement). Last but not the least, as mentioned before, cyto-STAR7 construct is not a suitable control as it is completely unrelated to mitochondrial translocation.*

We agree with the reviewer that the connection of CoQ to ferroptosis is well established. Here, we built on this exciting discovery and used CoQ-dependent ferroptotic resistance as a cellular readout to monitor CoQ transport to the plasma membrane.

Expressing STAR7 specifically either in the IMS or in the cytosol, we were able to unambiguously detangle the effect of STAR7 on mitochondrial function from its role for CoQ distribution. We have previously shown that expression of mito-STAR7 in *STAR7*^{-/-} cells is sufficient to preserve mitochondrial PC levels, cristae morphogenesis and OXPHOS (Saita et al., 2018). Here, we unravel a novel function of cytosolic STAR7 and demonstrate that it promotes transport of CoQ to the

plasma membrane. We feel that it is the strength of our approach that we can exclude any indirect effect of mitochondrial deficiencies or defects in the synthesis of CoQ. It should also be noted that the analysis of *Parl*^{-/-} mice demonstrated that the CoQ levels decrease in the brain before respiratory deficiencies, excluding indirect effects.

To comply with the request of the reviewer, we have now also extended our analysis to human colorectal cancer cells (HCT116). We isolated monoclonal *PARL*^{-/-} and *STARD7*^{-/-} HCT116 cells and monitored CoQ levels and ferroptotic resistance. As shown in Fig. 2e-h, loss of either PARL or STARD7 decreased CoQ levels and increased the sensitivity of the cells to erastin-induced ferroptosis, substantiating our findings in HeLa cells. We agree with the reviewer that CoQ synthesis and distribution is likely regulated in a tissue-specific manner, but to address this would require an extensive analysis of various tissues, which is outside the scope of the present manuscript.

As explained in detail under point 2, we do not propose that CoQ transport from mitochondria is coupled to the release of STARD7 from mitochondria upon PARL cleavage. Rather, our findings demonstrate that the presence of STARD7 in the cytosol is a prerequisite for CoQ distribution to the plasma membrane.

Minor points:

1. For the cell death analysis, why sometimes use cell death area/phase area, sometimes dead cell count/phase area?

We have corrected this and show now only cell death area/phase area in the cell death analysis. This does not affect any conclusion.

2. Why use Erastin + PUFA to induce ferroptosis? This condition is not common. More common inducing conditions are needed.

We used in early experiments a combination of erastin and PUFA to induce lipid peroxidation predominantly in the plasma membrane. However, it turned out that PUFA treatment is not essential. We now show in the manuscript that cyto-STARD7 overexpression protects against erastin, erastin and PUFA, and against RSL3-induced ferroptosis (Fig. 3). Moreover, we show that both *PARL*^{-/-} and *STARD7*^{-/-} cells are also more susceptible to ferroptosis induced by RSL3 (Fig. 2d) and erastin (Fig. 2g, h).

3. To confirm CoQ is required for STARD7's ferroptosis suppression function, direct CoQ synthesis inhibition (e.g. CoQ2 KO, 4-CBA) should be tested, in addition to iFSP1.

To address this point, we have used different concentrations of the COQ2 inhibitor 4-CBA to inhibit CoQ synthesis. Metabolomic experiments demonstrated that 4-CBA

treatment decreased CoQ levels by ~50% decrease (Fig. 3h, i). Similar to iFSP1, inhibition of CoQ synthesis abolished the protective effect of cyto-STARD7 against ferroptosis (Fig. 3j). This data further substantiates the requirement of CoQ for STARD7-mediated suppression of ferroptosis.

Reply to Reviewer #2:

This is a strong paper that ties together protein processing, CoQ transport and ferroptosis to generate a nice story that is supported by the data. I have mainly minor comments.

We thank the reviewer for this positive evaluation of our work and have carefully addressed the remaining concerns:

1 The focus of the paper is on the link between CoQ distribution within the cell and ferroptosis. While STARD7 is clearly involved in CoQ transport, the mechanism remains obscure. The authors discuss the difficulty of extracting such a hydrophobic molecule from the inner membrane, getting it from the mito IM and then to the mito OM and then -rightly in my view – suggest that extracting it to a transport protein and reinsert it into the plasma membrane is likely to require an extensive protein machinery. The authors could expand on this – there have been a few attempts to identify the putative machinery that moves CoQ from mitos to the plasma membrane (eg FEBS Journal (2010) 277 2067-2082. Free Radical Biology and Medicine (2020) 154 105-118).

The authors of the cited previous publications performed complementation studies in *coq2* yeast mutant cells using CoQ derivatives. These experiments indicated that the endocytic machinery might be required for the uptake of exogenous CoQ. We refer now to this work in the discussion.

To provide further insight into the mechanisms how STARD7 affects CoQ distribution, we have examined a possible direct interaction of STARD7 with CoQ (see also reply to Rev#1, point #1). Purified STARD7 was incubated with liposomes of varying phospholipid compositions containing CoQ4, CoQ9 and CoQ10. We then isolated STARD7 and analyzed associated lipids by mass spectrometry (Fig. 5). This approach revealed direct binding of CoQ4 to STARD7, demonstrating that STARD7 directly affects CoQ distribution. We did not observe CoQ9 and CoQ10 binding to STARD7. This likely relates to their hydrophobicity, which may prevent extraction from liposomes or recovery upon STARD7 purification. Interestingly, CoQ binding to STARD7 was only observed in the absence of DOPC in the liposomes, indicating competition between both STARD7-binding lipids (as also speculated by reviewer #3). Increasing the concentration of DOPC decreased CoQ4 binding to STARD7, highlighting the specificity of this interaction (Fig. 5c). The specificity of CoQ4 binding was further corroborated by mutating R189, which is located in the binding cavity of STARD7 and required for PC binding to STARD7 (Extended Fig. 5c; Bockelmann et al., 2018). The mutation R189Q abolished PC binding and strongly reduced CoQ4

binding (Fig. 5d), providing additional support for the specificity of CoQ4 binding to STARD7. Although the likely complex machinery involved in CoQ distribution deserves further experimentation in the future, our results demonstrate that STARD7 binds CoQ4 and thus directly affects CoQ transport.

2 The measurements of CoQ9/10 distribution between the mito and plasma membrane is not easy, but the data are presented as fold changes, which may have been done because of the high variability I'd expect in such an analysis. I really would like to see the actual amounts measured in the compartments normalised to protein or some other marker. It would also be good to measure the whole cell CoQ9/10 content and to see what proportion of the total pool is in the mito and plasma membranes.

We now show the raw data for CoQ levels in wildtype and cyto-STARD7 overexpressing cells in fraction 1 (8,000xg pellet) and fraction 3 (40,000xg pellet) in the Extended Fig. 4 (n=4). We also show now that the majority of total cellular CoQ is present in mitochondrial membranes (~90%), as expected (Fig. 4d, e). The amount of CoQ was normalized to the protein mass.

3 The mechanism of the selectivity for loss of CoQ biosynthesis machinery in the IM but not of OX phos was intriguing. The implication is that this is secondary to PC transport – was it confirmed that expression of mito-STARD7 restored mitochondrial IM PC levels? Also, did loss of STARD7 affect cristae morphology and was this reversed by mito-STARD7? Can you replicate this by altering PC synthesis?

We have previously demonstrated that expression of mito-STARD7 in *STARD7*^{-/-} cells was sufficient to restore PC levels, OXPHOS and cristae morphogenesis (Saita et al., 2018). We did not attempt to alter PC synthesis, since decreased PC levels cause pleiotropic effects (for instance impaired cristae structure and OXPHOS), hampering any interpretation. We feel that it is the strength of our approach that we are able to analyze the function of cytosolic STARD7 for CoQ transport independent of any indirect effects of defective mitochondria or deficiencies in the synthesis of PC or CoQ.

4 The authors inferred in Fig 5 that overexpression of cyto-STARD7 affected respiration by extracting CoQ from the IM. I couldn't see that loss of CoQ was actually measured and whether other interpretations are possible? Did cyto-STARD7 overexpression affect the CoQ biosynthetic machinery or OX PHOS complexes? Could overexpression of cyto-STARD7 have disrupted/sequestered PC and thereby indirectly affected mitochondria? in the paper and the authors might want to consider strengthening or removing it.

We have indeed measured CoQ10/9 levels in wildtype and cyto-STARD7 overexpressing cells after cell fractionation (see Fig. 4g). We show that overexpression of cyto-STARD7 decreased CoQ levels in the mitochondrial fraction

(fraction 1), while CoQ levels in the plasma membrane fraction were increased in (fraction 3). Decreased mitochondrial CoQ levels mildly affected OCR in cells grown on glucose and strongly limited OCR and cell growth on galactose medium (see Fig. 6a-d).

We have now also determined the cellular proteome of wildtype and cyto-STARD7 overexpressing cells. This analysis demonstrated that the steady state level of proteins of the CoQ synthesis pathway or of OXPHOS proteins are not decreased, excluding indirect effects (Extended Data Fig. 6a).

Minor points

1 Fig 1g shows cleavage of STAD7 by PARL with the whole cleaved protein temporarily in the IMS. Fig5e shows this more accurately with cleavage at the IM by PARL while the cytosolic module is still in the cytosol, precluding the need for a mysterious mechanism for the cleaved STAD7 to leave the IMS, implied in Fig 1g. I suggest replace Fig 1g with something more like Fig 5e.

We have optimized our model in Fig. 1g, which summarizes that PARL cleavage of STARD7 during mitochondrial import allows partitioning of STARD7 to both the IMS and the cytosol.

Reply to Reviewer #3:

In this article the authors discover a connection between the proteolytic activity of PARL, a mitochondrial rhomboid protease and the functional role of one of its polypeptide substrates, STARD7 in controlling transport of coenzyme Q (CoQ or ubiquinone). The authors show that PARL regulates both the synthesis and the cellular distribution of CoQ via its ability to proteolytically process STARD7, which changes its “zip code”, and relocates STARD7 from the mitochondria to the cytosol. Although PARL was previously known to affect the content of CoQ, the mechanism(s) responsible remained mysterious. This work unmasks the actions of PARL that affect both CoQ synthesis within the mitochondria, as well as the transport of this aqueous insoluble lipid to non-mitochondrial destinations within the cell. Transport of CoQ from mitochondria to the plasma membrane plays an important role in protecting cells from cell death due to lipid peroxidation, termed ferroptosis.

I believe that the paper could be enhanced if the authors addressed the following points:

1. The authors construct variants of STARD7 that accumulate exclusively in the mitochondrial ims (mito-STARD7) or in the cytosol (cyto-STARD7, that lacks the mitochondrial targeting sequence). However, it is important for the authors to determine experimentally that the said STARD7 variants are in fact exclusively located within these distinct compartments.

We have confirmed the localization of mito-STARD7 and cyto-STARD7 to the mitochondria and cytosol, respectively, by cell fractionation in our previous publication (Saita et al., 2018). We have substantiated these findings now also with immunofluorescence experiments of cyto-STARD7 overexpressing cells which were generated in this study (Extended Data Fig. 3b). For the sake of clarity, we now refer explicitly to these experiments in the manuscript.

2. Additionally, it might be interesting to express STARD7 that is targeted to the Mito inner membrane, but that is not able to be cleaved by PARL, as this would presumably prevent any retrograde relocation of processed STARD7 from the ims to the cytosol. This experiment would also address the author's claim that unprocessed STARD7 is present in the inner membrane of PARL^{-/-} cells but is not sufficient to ensure the accumulation of CoQ, as stated on page 6.

We have localized unprocessed STARD7 to the mitochondrial IMS in *PARL^{-/-}* cells in our original study, focusing on the role of PARL driving the dual localization of STARD7 (Fig. 2 in Saita et al. 2018). Since this work has already established the requirement of PARL-dependent STARD7 cleavage for the accumulation of STARD7 in the cytosol, we did not generate a variant of STARD7 that cannot be processed by PARL and therefore cannot be released to the cytosol (which would be also a challenging task given that the cleavage motif of PARL remains ill-defined). The analysis of *STARD7^{-/-}* cells expressing only mito-STARD7 unambiguously demonstrates that cyto-STARD7 is required for CoQ transport to the plasma membrane and ferroptosis.

3. In Figure 1j (and Supplementary Figure 1g) cells expressing the mutant STARD7R189Q appear to have a decreased content of CoQ10 (and CoQ9) relative to the STARD7 null mutant. This result seems surprising – that is, why would the point mutant have even lower CoQ content than the null mutant? The STARD7R189Q mutant is known to affect the binding of phosphatidyl choline (PC). Presumably the null mutant would also have a defect in binding PC. The authors conclude that maintenance of CoQ levels depend on the PC transport. Yet the result suggests that expression of STARD7R189Q has a more negative effect on CoQ content than does the null. Could it be that the STARD7R189Q mutant polypeptide has acquired an additional activity or a type of dominant negative effect that further decreases CoQ content relative to the null mutant?

We agree with the reviewer that the decreased CoQ content in *STARD7^{-/-}* cells expressing STARD7-R189Q is puzzling. We cannot exclude that this mutant acquired another function as speculated by the reviewer, but would rather favor a dominant effect of the variant. To examine this possibility, we attempted to express STARD7-R189Q in wildtype cells but could only obtain low expression levels. While this is in principle consistent with an inhibitory effect of this variant on cell growth, other possibilities cannot be excluded. Further experiments aiming to identify proteins binding to STARD7-R189Q but not STARD7 might help to explain the unexpectedly strong effect on CoQ levels. However, since this does not affect any

conclusion of the manuscript, we prefer to analyze this thoroughly in future experiments.

5. The authors convincingly demonstrate that suppression of ferroptosis (death by lipid peroxidation) depends on both mitochondrial and cytosolic STARD7, and they conclude that suppression of ferroptosis depends on STARD7 being present in both the IMS and the cytosol. The text in this section emphasizes that the addition of PUFA (arachidonic acid) would be expected to induce the lipid peroxidation in the plasma membrane (and so lead to cell death). However, it seems likely that addition of PUFA will impose lipid peroxidation on all intracellular membranes, including the mitochondrial membranes. In fact, recent literature (doi: 10.1038/s41586-021-03539-7.) suggests that the action of mitochondrial dihydroorotate dehydrogenase protects cells against PUFA-mediated ferroptosis via its ability to convert mitochondrial CoQ to CoQH2. The results in the present manuscript support this interpretation, and suggest that CoQH2 plays an important role as a chain breaking antioxidant in mitochondrial as well as the plasma membrane. Perhaps this could at least partially explain why the suppression of ferroptosis depends on both the mitochondrial and cytosolic forms of STARD7.

We agree with the reviewer that PUFAs likely foster lipid peroxidation in other cellular membranes as well and have changed the text accordingly (p. 8). Recent work has demonstrated the protective effect of DHODH and CoQ in the inner membrane of mitochondria against ferroptosis (Mao et al., 2021). As suggested by the reviewer, it is an attractive possibility that mito-STARD7 affects this pathway at least partially explaining its requirement for ferroptosis suppression.

6. Fig 5 depicts a model for the action of STARD7. Yet, the “squiggle” representing STARD7 is not terribly illuminating as to its mode of action. Is it possible that the START domain present in STARD7 might be able to bind CoQ as a ligand? The authors state on page 13 that, “...a complete membrane extraction of CoQ and binding with the lipid binding cavity of STARD7 appears unlikely give steric constraints and the hydrophobicity of the polyprenoid tail of CoQ.” What is the basis for this statement? Has modeling of CoQ within the hydrophobic START domain of STARD7 been attempted? STARD7 has a hydrophobic pocket that binds PC... is it possible that this pocket could “trade PC” for CoQ? This is how alpha-tocopherol transfer protein is postulated to interact with phosphatidylinositols and traffic vitamin E, so there is at least precedent for the idea that a phospholipid may “direct” the trafficking of lipids containing polyisoprenoid-based moieties.

We thank the reviewer for this insightful comment, which prompted us to examine a possible competition between PC and CoQ for STARD7 binding (see also response to Rev#1 point#1 and Rev#2 point#1).

Purified STARD7 was incubated with liposomes of varying phospholipid compositions containing CoQ4, CoQ9 and CoQ10. We then isolated STARD7 and analyzed associated lipids by mass spectrometry (Fig.5). This approach revealed direct binding of CoQ4 to STARD7. Interestingly, CoQ binding to STARD7 was only observed in the absence of DOPC in the liposomes, indicating competition between both STARD7-binding lipids. Increasing the concentration of DOPC decreased CoQ4 binding to STARD7, highlighting the specificity of this interaction (Fig. 5c). The

specificity of CoQ4 binding was further corroborated by mutating R189, which is located in the binding cavity of STARD7 and required for PC binding to STARD7 (Extended Fig. 5c). The mutation R189Q also impaired CoQ4 binding (Fig. 5d), providing additional support for the specificity of the observed interaction.

Together these experiments demonstrate binding of CoQ to STARD7 *in vitro* and competition between CoQ and PC binding.

It should be noted that we did not observe binding of CoQ9/10 to STARD7 *in vitro*. This likely relates to their hydrophobicity, which may prevent their extraction from liposomes or their recovery upon STARD7 purification. The extreme hydrophobicity of CoQ9/10 is also the basis for our speculation that complete membrane extraction appears unlikely. Rather, a complex machinery might be involved in CoQ distribution which deserves further experimentation in the future.

General comments

1. It can be confusing to refer to “reduced” levels of CoQ. This is because reduced can mean either decreased, or alternatively, can indicate that the CoQ is reduced (e.g. CoQ is reduced to the CoQH2 hydroquinone). It would be helpful for the authors to peruse their text for this usage, and to replace the word “reduced” with “decreased” whenever the intent is to indicate a decreased content of CoQ.

We apologize for the confusing wording and have corrected this in the revised manuscript.

2. There are two isoforms of human STARD7, STARD7-I and STARD7-II. How do these two isoforms relate to the PARL mediated processing reported here? This could be important since the processing by PARL may differ depending on the isoform being expressed.

Previous reports refer to the precursor and mature form of STARD7 as STARD7-I and STARD7-II. STARD7-II does not contain the MTS and PARL cleavage site (Horibata and Sugimoto, 2010) and likely represents the mature form of STARD7 generated by PARL cleavage. Consistently, only one STARD7 form accumulates quantitatively in mitochondria of *PARL*^{-/-} cells (Saita et al., 2018).

Decision Letter, first revision:

Our ref: NCB-A48412A

10th October 2022

Dear Thomas,

Thank you for submitting your revised manuscript "Mitochondria regulate intracellular coenzyme Q transport and ferroptotic resistance via STARD7" (NCB-A48412A). It has now been seen by two of the original referees and their comments are below. The reviewers find that the paper has improved in revision. Reviewer #3 was unavailable to re-review; however, given the overlap in comments across reviewers, we asked both Rev#1 and Rev#2 to please weigh in on how you addressed Rev#3's comments. Both Revs#1-2 kindly agreed and shared in comments to the editors that the revisions were strong and addressed the core issues of Rev#3.

Therefore, we'll be happy in principle to publish the study in Nature Cell Biology, pending minor revisions to comply with our editorial and formatting guidelines.

Please note that the current version of your manuscript is in a PDF format. Could you please email us a copy of the file in an editable format (Microsoft Word or LaTeX)? We cannot proceed with PDFs at this stage. Many thanks for your attention to this point.

With the Word file, we will be performing detailed checks on your paper and will send you a checklist detailing our editorial and formatting requirements within 1-2 weeks of the start of the checks. Please do not upload the final materials and make any revisions until you receive this additional information from us.

Thank you again for your interest in Nature Cell Biology. Please do not hesitate to contact me if you have any questions.

Sincerely,

Melina

Melina Casadio, PhD
Senior Editor, Nature Cell Biology
ORCID ID: <https://orcid.org/0000-0003-2389-2243>

Reviewer #1 (Remarks to the Author):

The authors have satisfactorily addressed major concerns from this reviewer.

Reviewer #2 (Remarks to the Author):

None

Decision Letter, final checks:

Our ref: NCB-A48412A

26th October 2022

Dear Dr. Langer,

Thank you for your patience as we've prepared the guidelines for final submission of your Nature Cell Biology manuscript, "Mitochondria regulate intracellular coenzyme Q transport and ferroptotic resistance via STARD7" (NCB-A48412A). Please carefully follow the step-by-step instructions provided in the attached file, and add a response in each row of the table to indicate the changes that you have made. Please also check and comment on any additional marked-up edits we have proposed within the text. Ensuring that each point is addressed will help to ensure that your revised manuscript can be swiftly handed over to our production team.

In recognition of the time and expertise our reviewers provide to Nature Cell Biology's editorial process, we would like to formally acknowledge their contribution to the external peer review of your manuscript entitled "Mitochondria regulate intracellular coenzyme Q transport and ferroptotic resistance via STARD7". For those reviewers who give their assent, we will be publishing their names alongside the published article.

Nature Cell Biology offers a Transparent Peer Review option for new original research manuscripts submitted after December 1st, 2019. As part of this initiative, we encourage our authors to support increased transparency into the peer review process by agreeing to have the reviewer comments, author rebuttal letters, and editorial decision letters published as a Supplementary item. When you submit your final files please clearly state in your cover letter whether or not you would like to participate in this initiative. Please note that failure to state your preference will result in delays in accepting your manuscript for publication.

Cover suggestions

As you prepare your final files we encourage you to consider whether you have any images or illustrations that may be appropriate for use on the cover of Nature Cell Biology.

Nature Cell Biology has now transitioned to a unified Rights Collection system which will allow our Author Services team to quickly and easily collect the rights and permissions required to publish your work. Approximately 10 days after your paper is formally accepted, you will receive an email in providing you with a link to complete the grant of rights. If your paper is eligible for Open Access, our Author Services team will also be in touch regarding any additional information that may be required to arrange payment for your article.

Please note that *Nature Cell Biology* is a Transformative Journal (TJ). Authors may publish their research with us through the traditional subscription access route or make their paper immediately open access through payment of an article-processing charge (APC). Authors will not be required to make a final decision about access to their article until it has been accepted. Find out more about Transformative Journals

Please use the following link for uploading these materials:
[REDACTED]

Best regards,

Kendra Donahue
Staff
Nature Cell Biology

On behalf of

Melina Casadio, PhD
Senior Editor, Nature Cell Biology
ORCID ID: <https://orcid.org/0000-0003-2389-2243>

Reviewer #1:

Remarks to the Author:

The authors have satisfactorily addressed major concerns from this reviewer.

Reviewer #2:

Remarks to the Author:

None

Final Decision Letter:

Dear Dr Langer,

I am pleased to inform you that your manuscript, "Mitochondria regulate intracellular coenzyme Q transport and ferroptotic resistance via STARD7", has now been accepted for publication in Nature Cell Biology. Congratulations on this beautiful study!

Please note that *Nature Cell Biology* is a Transformative Journal (TJ). Authors may publish their research with us through the traditional subscription access route or make their paper immediately open access through payment of an article-processing charge (APC). Authors will not be required to make a final decision about access to their article until it has been accepted. Find out more about Transformative Journals

If you have not already done so, we strongly recommend that you upload the step-by-step protocols used in this manuscript to the Protocol Exchange (www.nature.com/protocolexchange), an open online resource established by Nature Protocols that allows researchers to share their detailed experimental know-how. All uploaded protocols are made freely available, assigned DOIs for ease of citation and are fully searchable through nature.com. Protocols and Nature Portfolio journal papers in which they are used can be linked to one another, and this link is clearly and prominently visible in the online versions of both papers. Authors who performed the specific experiments can act as primary authors for the Protocol as they will be best placed to share the methodology details, but the Corresponding Author of the present research paper should be included as one of the authors. By uploading your Protocols to Protocol Exchange, you are enabling researchers to more readily reproduce or adapt the methodology you use, as well as increasing the visibility of your protocols and papers. You can also establish a dedicated page to collect your lab Protocols. Further information can be found at www.nature.com/protocolexchange/about

With kind regards,

Melina

Melina Casadio, PhD
Senior Editor, Nature Cell Biology
ORCID ID: <https://orcid.org/0000-0003-2389-2243>

** Visit the Springer Nature Editorial and Publishing website at www.springernature.com/editorial-and-publishing-jobs for more information about our career opportunities. If you have any questions please click here.**